# An Evolutionary Algorithm for Black-Box Adversarial Attack Against Explainable Methods

**Phoenix Williams** *phoenix.n.williams@gsk.com*
*GSK.ai*

**Jessica Schouff** *jessica.v.schrouff@gsk.com*
*GSK.ai*

**Lea Goetz** *lea.x.goetz@gsk.com*
*GSK.ai*

**Reviewed on OpenReview:** *https://openreview.net/forum?id=MlUP5Euj6S*

## Abstract

The explainability of deep neural networks (DNNs) remains a major challenge in developing trustworthy AI, particularly in high-stakes domains such as medical imaging. Although explainable AI (XAI) techniques have advanced, they remain vulnerable to adversarial perturbations, underscoring the need for more robust evaluation frameworks. Existing adversarial attacks often focus on specific explanation strategies, while recent research has introduced black-box attacks capable of targeting multiple XAI methods. However, these approaches typically craft pixel-level perturbations that require a large number of queries and struggle to effectively attack less granular XAI methods such as Grad-CAM and LIME. To overcome these limitations, we propose a novel attack that generates perturbations using semi-transparent, RGB-valued circles optimized via an evolutionary strategy. This design reduces the number of tunable parameters, improves attack efficiency, and is adaptable to XAI methods with varying levels of granularity. Extensive experiments on medical and natural image datasets demonstrate that our method outperforms state-of-the-art techniques, exposing critical vulnerabilities in current XAI systems and highlighting the need for more robust interpretability frameworks.

## 1 Introduction

Deep neural networks (DNNs) have revolutionized the field of computer vision, driving significant advancements across a variety of tasks Lin et al. (2014); Simonyan & Zisserman (2015); Springenberg et al. (2015). In healthcare, artificial intelligence is becoming a transformative force, offering groundbreaking solutions for diagnosis, treatment, and patient care Chaddad et al. (2023). Yet, the black-box nature of many DNNs raises concerns regarding their explainability, accountability, and trustworthiness Quinn et al. (2021); Rane et al. (2023); Rosenbacke et al. (2024b). To address these issues and bolster trust, explainable artificial intelligence (XAI) has emerged as a pivotal area of research. By understanding the decision-making processes of complex DNNs, XAI fosters confidence among healthcare providers and patients Dosilovic et al. (2018). Within the computer vision domain, explanation methods frequently generate attribution maps that visualize feature importance, illustrating how different elements on an image contribute to a DNN's predictions Simonyan et al. (2014); Shrikumar et al. (2017); Selvaraju et al. (2017); Lundberg & Lee (2017); Böhle et al. (2024).

Despite the advancements of XAI, recent studies have revealed that many existing methods remain vulnerable to adversarial inputs Tamam et al. (2023); Huang et al. (2023); Baniecki & Biecek (2024). Such inputs, generated through imperceptible perturbations (as illustrated in Figure 1), have shown the ability to simultaneously alter both the XAI attribution maps and the classification outputs of DNNs Huang et al.

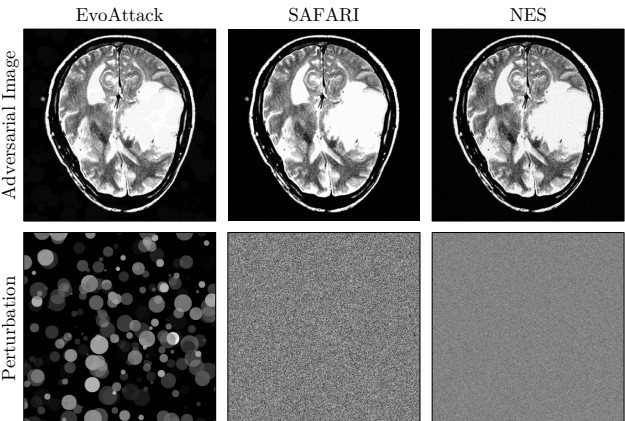

Figure 1: Adversarial images and perturbations generated by the NES Tamam et al. (2023), SAFARI Huang et al. (2023) and the proposed EvoAttack algorithm when attacking an image from the Br35h dataset. Adversarial perturbations generated by NES and SAFARI perturb every pixel of the image whereas the perturbation generated by the proposed EvoAttack method is constructed using a set of 300 *circle* shapes

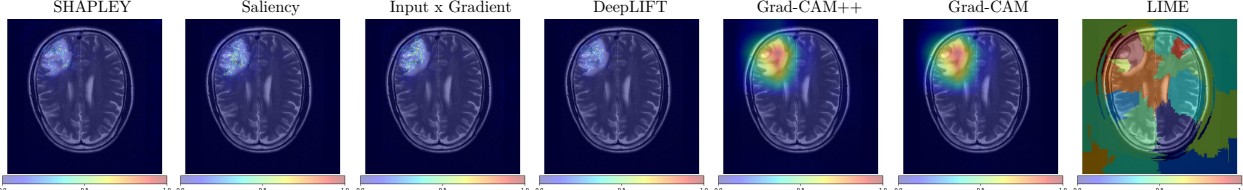

Figure 2: Attribution maps generated by XAI methods. These methods are applied to explain a 'true' tumour classification of an image from the Br35h dataset made by a trained VGG-16 classifier. We observe that DeepLIFT, SHAPLEY, Saliency and Input x Gradient methods produce attribution maps with high-granularity, emphasizing important pixels. In contrast, Grad-CAM, Grad-CAM++ and LIME generate attribution maps that capture more global features, highlighting broader regions of the image.

(2023). The presence of these adversarial examples in real-world settings is particularly concerning in domains where DNN explainability is essential or legally required, such as autonomous driving Omeiza et al. (2022) and healthcare Chaddad et al. (2023); Hao et al. (2024); van der Velden et al. (2022). Consequently, the development of adversarial attack techniques has become a crucial research direction for evaluating and improving the robustness of XAI methods Tamam et al. (2023); Huang et al. (2023).

Early research efforts primarily focused on crafting adversarial images by targeting specific XAI methods and leveraging knowledge of the underlying DNN architecture and parameters, a setting known as white-box attacks Wang et al. (2023); Moosavi-Dezfooli et al. (2016); Zhang et al. (2020); Ghorbani et al. (2019). However, because these approaches depend on access to internal DNN information, they often fail to generalize across different explanation techniques. Consequently, recent work has shifted toward the black-box scenario, where only input–output pairs from the DNN and the XAI method are accessible Tamam et al. (2023); Huang et al. (2023). In this setting, most existing attacks employ meta-heuristic approaches Tamam et al. (2023); Huang et al. (2023) inspired by evolutionary algorithms Li et al. (2024).

While existing methods have successfully generated adversarial images against XAI techniques, they face key limitations. Firstly, these methods often require extensive querying of both the DNN and the XAI method to achieve meaningful distortions in attribution maps. This dependency poses substantial challenges in environments where query budgets are limited or expensive, whether due to financial constraints Ilyas et al. (2018); Dhabliya et al. (2024) or time restrictions Keddous et al. (2023). As a result, conducting robustness evaluations that involve adversarial attacks with high query budgets becomes costly, impacting both financial resources and development time. This issue stems from the use of population-based approaches

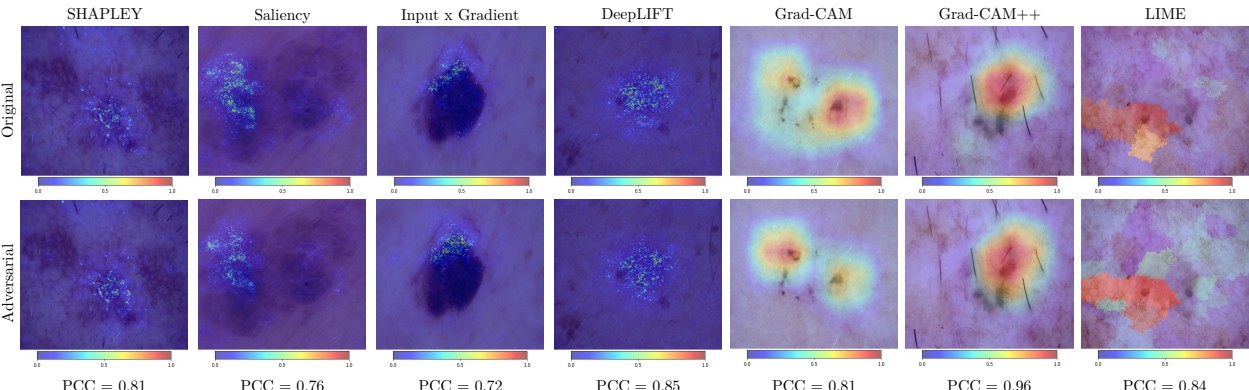

Figure 3: Figure shows adversarial images produced by the EvoAttack method when applied to the Task 1 scenario (misclassification with preserved explanation), along with the respective generated attribution maps. Both original and adversarial explanations on the HAM10000 images are visually similar with PCC values of 0.7 and above. Both sets of explanations highlight seemingly relevant regions of the image, however, all adversarial images cause the underlying VGG-16 DNN to misclassify.

combined with the inherently high-dimensional nature of the search space—for instance, attacking an image from the HAM10000 dataset Tschandl et al. (2018) with dimensions $(450 \times 600 \times 3)$ results in searching through a space of $810,000$ dimensions. Secondly, existing attacks often overlook the varying granularity of XAI methods' explanations when designing perturbations, as shown in Figure 2. Current approaches tend to modify all pixels independently, which is effective when targeting XAI methods that produce detailed, pixel-level explanation maps, such as SHAPLEY or Saliency methods. However, these approaches struggle against XAI methods that emphasize broader, global regions, leading to increased robustness in methods like Grad-CAM and Grad-CAM++ Huang et al. (2023). This oversight highlights the need for more adaptive attack strategies that consider the explanatory granularity of different XAI techniques.

To address these limitations, we propose a novel attack method inspired by image approximation techniques from the computational art community Lambert et al. (2013); Garbaruk et al. (2022); Tian & Ha (2022). Our approach generates adversarial perturbations using a set of semi-transparent, RGB-valued circles whose parameters are optimized through an evolutionary strategy. Constructing perturbations in this manner substantially reduces the search space that remains consistent across images of varying sizes, thereby improving attack efficiency. Moreover, by constraining the size of the shapes, our attack can effectively target XAI method that produce attribution maps with different levels of granularity.

The remainder of this paper is organized as follows: Section 2 reviews related works, highlighting key contributions and limitations. Section 3 introduces the proposed attack and provides a detailed explanation of its implementation. Section 4 presents the evaluation metrics and experimental results, followed by a comprehensive analysis. Finally, Section 5 concludes the paper and discusses potential directions for future research.

## 2 Related Works

Adversarial attacks on DNNs have become one of the most active research areas within the machine learning community, predominantly focusing on targeting DNN classifiers Williams & Li (2023b); Dong et al. (2025); Ilyas et al. (2018); Williams & Li (2023a); Madry et al. (2017); Andriushchenko et al. (2020). More recently, there has been increasing interest in exploring the effects of adversarial perturbations on attribution maps produced by XAI methods, addressing both white-box Heo et al. (2019); Moosavi-Dezfooli et al. (2016); Zhang et al. (2020); Ghorbani et al. (2019); Kindermans et al. (2019); Subramanya et al. (2019); Dombrowski et al. (2019); Kuppa & Le-Khac (2020) and black-box Tamam et al. (2023); Huang & Zhang (2020) settings. Our work situates itself within this domain by focusing on the black-box scenario. Unlike existing black-box approaches that generate pixel-wise perturbations, our method employs semi-transparent circular shapes,

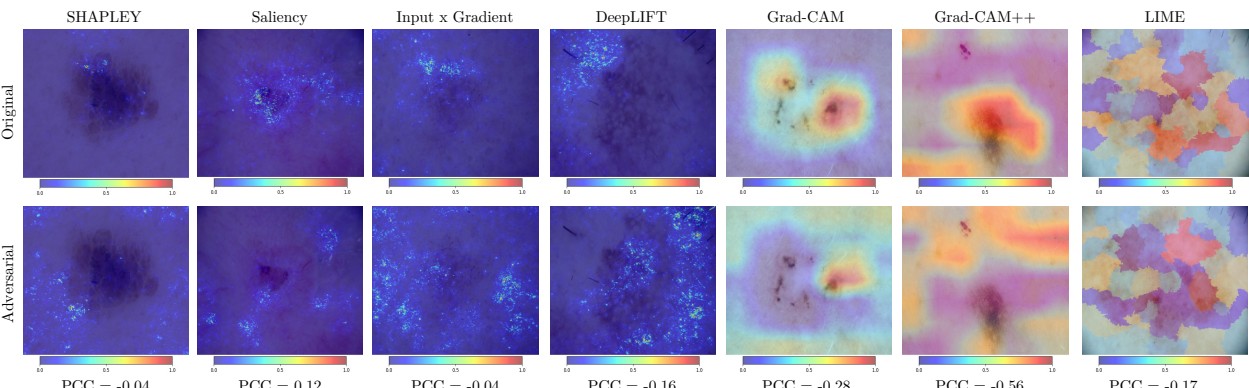

Figure 4: Figure shows adversarial images produced by the EvoAttack method when applied to the Task 2 scenario (correct classification with distorted explanation), along with the respective generated attribution maps. The original and adversarial explanations on the HAM10000 images are visually dissimilar with PCC values of −0.04 and below. Despite the distortion, all adversarial images are correctly classified by the VGG-16 DNN.

which substantially reduces the number of optimizable parameters and can be easily adapted to various XAI methods, addressing key limitations of current strategies.

As DNNs become increasingly deployed within real-world applications, ensuring their decision-making processes are interpretable is essential Doshi-Velez et al. (2017), underscoring the importance of developing robust and accurate explanations. Adversarial attacks against XAI methods are designed to assess the sensitivity of explanations to minor changes in input images Alvarez-Melis & Jaakkola (2018). While traditional adversarial attacks on DNNs focus on inducing misclassification, attacks against XAI methods aim to challenge the reliability of the provided explanations by either 1) causing a minimal distortion to an attribution map whilst causing misclassification Huang et al. (2023), or 2) maximizing the distortion of an attribution map whilst maintaining a correct classification Tamam et al. (2023) (shown in Figures 3 and 4).

In their pioneering work, Ghorbani et al. introduced the concept of adversarial attacks on XAI methods by targeting gradient-based feature attributions of convolutional DNNs. The authors iteratively perturbed inputs in the direction that altered the explanation's gradient. Concurrently, research by Kindermans et al. highlighted the sensitivity of explanations to slight input transformations, although they did not directly propose methods for constructing such attacks. Subsequent works exploiting gradient information have constructed adversarial images by altering all pixels in the image Zhang et al. (2020) in addition to localised regions (patches) Selvaraju et al. (2017).

Whilst most existing methods focusing on the white-box setting, recent efforts have shifted toward developing black-box approaches. In the black-box scenario, only the input-output pairs of the DNN and XAI methods are accessible, allowing these techniques to be applied to any explanation method that produces an attribution map Huang et al. (2023). Existing black-box techniques often employ heuristic optimization methods to craft an adversarial image by solving a constructed loss function. Tamam et al. utilize a Natural Evolutionary Strategy (NES) optimiser Wierstra et al. (2014) to minimize the loss function previously proposed in Dombrowski et al. (2019). This attack iteratively estimated the gradient of the loss function by computing the finite-differences between a set of points sampled from a normal distribution. Conversely, the attack method by Huang et al. does not rely on any gradient information. Instead, they adapt the POBA-GA genetic algorithm developed by Chen et al. to evolve a population of solutions through crossover and mutation genetic operators. The authors demonstrate the superior performance of their method by targeting both gradient-based and perturbation-based explanation methods.

Despite recent advances in black-box adversarial attacks against XAI methods, the large number of queries required for both the DNN and the XAI method raises concerns about their practicality in financially Dhabliya et al. (2024) or time-constrained Keddous et al. (2023) scenarios. Moreover, the extensive computational effort

needed to generate adversarial examples limits the realistic evaluation of explanation robustness Wu et al. (2021). In addition, existing approaches overlook the granularity of XAI explanations, typically perturbing input images at the pixel level. While such fine-grained modifications perform well against highly detailed methods such as saliency maps and DeepLIFT, they struggle to succeed when attacking XAI techniques that focus on less granular features, such as Grad-CAM.

For a comprehensive survey of adversarial attacks on XAI methods, readers are referred to Baniecki & Biecek (2024). A detailed discussion of black-box adversarial attacks against image classifiers is also provided in Appendix Section A.11.

## 3 Proposed Method

The goal of our method is to generate adversarial images that either (i) induce misclassification while minimizing distortion to the attribution map or (ii) maximise the distortion of the attribution map while preserving the model's original classification Huang et al. (2023), all under a constrained query budget. We design the perturbation as a set of semi-transparent, RGB-valued circles, thereby reducing the search-space dimensionality to the circles' attributes. Similar to existing approaches, we generate adversarial perturbations by optimizing a distance-based objective function. We begin this section by formulating the problem, followed by a detailed description of each component of our proposed method. The overall structure of our approach is summarized in Figure 6 within the appendix.

### 3.1 Problem Formulation

Consider a trained DNN classifier $f : \mathcal{X} \subseteq [0,1]^{h \times w \times 3} \to \mathbb{R}^P$ which takes a single benign RGB image we wish to attack $\mathbf{x} \in \mathcal{X}$ of height $h$ and width $w$, and outputs a label $y = \underset{p \in \{1, \cdots, P\}}{\text{argmax}} f_p(\mathbf{x})$, where $P$ is the total number of class labels. Further, let $g(\cdot, \cdot)$ be an explanation function, which takes a trained DNN $f$ and benign image $\mathbf{x}$ as inputs. In this work, we assume access to the output probabilities of the classifier $f$, and attack an XAI method $g$ that outputs an attribution map $g(\cdot, \cdot) \to \mathbb{R}^{h \times w}$, where its height and width match the input image $\mathbf{x}$.

To preserve the semantic integrity of the image, we adhere to existing attack methods by constraining the perturbation size using the $l_\infty$ norm Huang et al. (2023); Tamam et al. (2023); Dombrowski et al. (2019). Consequently, we aim to generate a perturbation $\boldsymbol{\delta}$ that solves the following optimization problems:

**Task 1:**

$$
\begin{aligned}
\underset{\boldsymbol{\delta}}{\text{minimize}} \quad & \mathcal{D}(g(\mathbf{x}), g(\mathbf{x} + \boldsymbol{\delta})) \\
\text{subject to} \quad & ||\boldsymbol{\delta}||_\infty \leq \epsilon, \\
& \mathcal{L}(f; \mathbf{x} + \boldsymbol{\delta}, y_q) < 0, \\
& 0 \leq \mathbf{x} + \boldsymbol{\delta} \leq 1
\end{aligned}
\tag{1}
$$

**Task 2:**

$$
\begin{aligned}
\underset{\boldsymbol{\delta}}{\text{maximize}} \quad & \mathcal{D}(g(\mathbf{x}), g(\mathbf{x} + \boldsymbol{\delta})) \\
\text{subject to} \quad & ||\boldsymbol{\delta}||_\infty \leq \epsilon, \\
& \mathcal{L}(f; \mathbf{x} + \boldsymbol{\delta}, y_q) > 0, \\
& 0 \leq \mathbf{x} + \boldsymbol{\delta} \leq 1,
\end{aligned}
\tag{2}
$$

where $\mathcal{D}$ represents the distance measure between the attribution maps produced for the benign and adversarial image and $\epsilon$ controls the extent of the perturbation's impact on the benign image. We follow the setup of Huang et al. by defining $\mathcal{D}$ as the $1/PCC$ where $PCC$ is the Pearson Correlation Coefficient. We ensure that the value of the loss function $\mathcal{L}(\cdot)$ is negative when $\mathbf{x} + \boldsymbol{\delta}$ results in misclassification by defining the loss in the constraint as the marginal loss:

$$
\mathcal{L}(f; \mathbf{x} + \boldsymbol{\delta}, y) = f_y - f_{y_q},
\tag{3}
$$

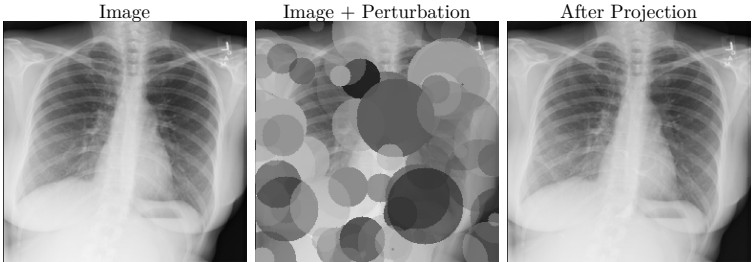

Figure 5: The process of repairing an adversarial image $\mathbf{x} + \boldsymbol{\delta}$ (described in equation (5)) to ensure it satisfies the constraints of (1) and (2), where $\epsilon = 6/255$

where $y$ corresponds to the true label of $\mathbf{x}$ and $y_q = \underset{q \neq y}{\operatorname{argmax}} \; f_p(\mathbf{x})$ is a label corresponding to a class other than the true label $y$.

### 3.2 Perturbation Initialization

To alleviate the issues associated with the large number of tunable values of an image perturbation, we construct a perturbation by overlaying $N$ RGB-valued semi-transparent circles, drawing inspiration from techniques prevalent within the computational art field Lambert et al. (2013); Garbaruk et al. (2022); Tian & Ha (2022).

In this work, we construct the adversarial perturbation $\boldsymbol{\delta}$ through the concatenation of $N$ shapes:

$$\boldsymbol{\delta} = \boldsymbol{\delta}^1 \oplus \boldsymbol{\delta}^2 \cdots \oplus \boldsymbol{\delta}^N \tag{4}$$

where $\oplus$ denotes the concatenation operator and $\boldsymbol{\delta}^a$ is the $a$-th shape applied to a blank array. In the proposed attack, each shape $\boldsymbol{\delta}^a$, for $a \in 1, \cdots, N$, is represented by a vector consisting of seven elements: the centre's coordinates $(c_1^a, c_2^a)$, the radius $r^a \times$ (Max Diameter) $\times (w \cdot h)$, where 'Max Diameter' is a tunable parameter controlling the size of the circles, and $w, h$ are the width and height of the attacked image, respectively. Finally, the vector includes the RGB values $R^a, G^a, B^a$, and the transparency $T^a$. These elements are normalized to continuous values between 0 and 1 and are initially sampled randomly from a uniform distribution, $\boldsymbol{\delta}^a \sim \mathcal{U}(0, 1)$. Constructing the perturbation in this manner reduces the number of optimization variables significantly—from $h \times w \times 3$ (which totals $150{,}528$ for a typical $(224, 224, 3)$ ImageNet image) to $N \times 7$, where $N$ is the number of circle shapes used to construct the perturbation defined by the user. Importantly, this perturbation construction method is also invariant to the image size.

### 3.3 Adversarial Image Construction

Given an adversarial perturbation $\boldsymbol{\delta}$ and a benign image $\mathbf{x}$, where both $\mathbf{x}$ and $\boldsymbol{\delta}$ belong to $\mathbb{R}^{h \times w \times 3}$, the corresponding adversarial image $\mathbf{x}^*$ is generated by equation (4) which overlaps all shapes from $\boldsymbol{\delta}$ onto $\mathbf{x}$. To ensure $\mathbf{x}^*$ complies with the constraints of (1) and (2), we project the pixels of the constructed adversarial image as follows:

$$\mathbf{x}_i^* = \begin{cases} \mathbf{x}_i + \epsilon & \text{if } \mathbf{x}_i^* > \mathbf{x}_i + \epsilon \\ \mathbf{x}_i - \epsilon & \text{if } \mathbf{x}_i^* < \mathbf{x} - \epsilon \\ \mathbf{x}_i^* & \text{otherwise,} \end{cases} \tag{5}$$

where $i$ is a pixel index. For greyscale images, we convert the RGB values of each circle to their greyscale equivalents before placing them onto the benign image. We visualize the projection process in Figure 5 which ensures the constraints of (1) and (2) are satisfied.

### 3.4 Perturbation Optimization

To optimize the properties of each shape $\boldsymbol{\delta}^a$, we utilize a single-solution evolutionary strategy known as $(1+1)$-ES. In each iteration, a child solution $\boldsymbol{\delta}^{**}$ is generated by stochastically modifying its parent $\boldsymbol{\delta}^*$ with

values sampled from a normal distribution $\sigma \cdot \mathcal{N}(0, I)$. The parameter $\sigma$ is adjustable, allowing a balance between exploration—searching unexplored regions of the solution space—and exploitation—fine-tuning the current solution. A larger $\sigma$ facilitates exploration, while a smaller value encourages exploitation. The child solution replaces the parent if it demonstrates superior performance on the task at hand.

Traditionally, the selection process is based solely on comparing the objective values of the parent and child solutions. However, due to the constraints outlined in equations (1) and (2), a more nuanced selection process is warranted. Huang et al. addressed this issue for (1) by proposing a heuristic method related to the dynamics of the evolutionary population, and resolved (2) by multiplying the objective value of solutions not satisfying the constraint with $-1$. Alternatively, Tamam et al. proposed a methodology that combines $\mathcal{D}$ and $\mathcal{L}$ using a weighted sum of its estimated gradients. Nonetheless, the single-solution architecture of our proposed method limits its ability to incorporate these existing strategies.

Instead, we draw inspiration from evolutionary computation research that addresses complex constrained optimization problems using dominance functions Coello Coello & Mezura-Montes (2002); Williams & Li (2023b); Williams et al. (2023), similar to techniques employed in the multi-objective optimization domain Deb (2001). For each task, a child solution $\boldsymbol{\delta}^{**}$ replaces its parent $\boldsymbol{\delta}^{*}$ if any of the following conditions are satisfied:

**Definition 1 (Task 1 Domination)** *:*

- $\mathcal{L}(\boldsymbol{\delta}^{*}) \geq 0$ *and* $\mathcal{L}(\boldsymbol{\delta}^{**}) < 0$ .

- *Both* $\mathcal{L}(\boldsymbol{\delta}^{*}) < 0$ *and* $\mathcal{L}(\boldsymbol{\delta}^{**}) < 0$ *and* $\mathcal{D}(g(\mathbf{x}), g(\mathbf{x} + \boldsymbol{\delta}^{**})) < \mathcal{D}(g(\mathbf{x}), g(\mathbf{x} + \boldsymbol{\delta}^{*}))$

- *Both* $\mathcal{L}(\boldsymbol{\delta}^{*}) > 0$ *and* $\mathcal{L}(\boldsymbol{\delta}^{*}) > 0$ *and* $\mathcal{L}(\boldsymbol{\delta}^{**}) < \mathcal{L}(\boldsymbol{\delta}^{*})$

**Definition 2 (Task 2 Domination)** *:*

- $\mathcal{L}(\boldsymbol{\delta}^{*}) \leq 0$ *and* $\mathcal{L}(\boldsymbol{\delta}^{*}) > 0$ .

- *Both* $\mathcal{L}(\boldsymbol{\delta}^{*}) > 0$ *and* $\mathcal{L}(\boldsymbol{\delta}^{*}) > 0$ *and* $\mathcal{D}(g(\mathbf{x}), g(\mathbf{x} + \boldsymbol{\delta}^{**})) > \mathcal{D}(g(\mathbf{x}), g(\mathbf{x} + \boldsymbol{\delta}^{*}))$

- *Both* $\mathcal{L}(\boldsymbol{\delta}^{*}) < 0$ *and* $\mathcal{L}(\boldsymbol{\delta}^{*}) < 0$ *and* $\mathcal{L}(\boldsymbol{\delta}^{**}) < \mathcal{L}(\boldsymbol{\delta}^{*})$.

These domination definitions are designed to guide our attack method in generating perturbation values that either minimize (for task 1) or maximize (for task 2) the distortion of the explanation, while simultaneously satisfying the respective classification constraints.

## 4 Experiments

Most existing studies assess their methods by attacking explanation techniques applied to ImageNet trained classifiers. In contrast, our work additionally focuses on classification tasks within the medical imaging domain, where the explainability of a DNN's decisions is critically important Chaddad et al. (2023); Hao et al. (2024); van der Velden et al. (2022). Therefore, we target explanation methods applied to DNN classifiers trained on three distinct medical image datasets and ImageNet. The experimental setup is outlined in Section 4.1, which is followed by a comparative analysis of leading attack methods SAFARI Huang et al. (2023), NES Tamam et al. (2023) and Square Attack Andriushchenko et al. (2020), detailed in Section 4.2. Subsequently, we utilise our proposed method to rank the robustness of various XAI techniques, as presented in Section 4.3. In Section 4.4, we evaluate XAI methods applied to adversarially trained models that incorporate adversarial images generated by our attack method within their training processes. Lastly, Section 4.5 presents an ablation study that examines the significance of different components and parameters of our proposed approach.

### 4.1 Experimental Setup

**Datasets:** We evaluate XAI methods on deep neural network (DNN) classifiers trained across three medical imaging datasets as well as ImageNet. The HAM10000 dataset Tschandl et al. (2018) contains approximately

$10,000$ dermatology images of pigmented skin lesions spanning seven diagnostic categories, including malignant cases. The Br35h dataset Hamada (2020) consists of $3,000$ brain MRI scans, annotated to indicate the presence or absence of tumours. The COVID-QU-Ex dataset Tahir et al. (2021) provides $33,920$ chest X-rays classified into three categories: COVID-19, pneumonia, and normal. Following prior work Huang et al. (2023); Tamam et al. (2023), we randomly sample 100 correctly classified images from each dataset's test or validation split to conduct adversarial attacks. All images are resized to $(224 \times 224 \times 3)$ prior to input into the DNN models.

**Explanation and Classifier Settings:** In this study, we explore seven XAI methods to assess their robustness and efficiency. Specifically, we evaluate Grad-CAM Selvaraju et al. (2017), Grad-CAM++ Chattopadhyay et al. (2017), Saliency Maps Simonyan et al. (2014); Selvaraju et al. (2017), DeepLIFT Shrikumar et al. (2017), Gradient x Input Shrikumar et al. (2017), LIME Peng & Menzies (2021), and Shapley Lundberg & Lee (2017). For the classifiers, we employ the architectures of MobileNet Howard et al. (2019), AlexNet Krizhevsky et al. (2012), and VGG-16 Simonyan & Zisserman (2015). Given the relatively small size of our medical datasets, we fine-tune models pre-trained on ImageNet using the PyTorch library Paszke et al. (2019). Each model undergoes fine-tuning over 10 epochs, with a batch size of 32 and a learning rate of $1 \times 10^{-4}$, utilizing the ADAM optimizer Kingma & Ba (2015) and cross entropy loss. The datasets are divided into training, validation, and testing subsets with a ratio of $70\%/10\%/20\%$. Detailed performance metrics are provided in Section A.2 in the appendix. All experiments were executed on an NVIDIA RTX A6000 GPU system.

**Parameter Settings:** To ensure the generated perturbations cause minimal semantic alterations to the attacked images, we set the value of $\epsilon$ using previous adversarial attack research targeting image classification DNNs Rusu et al. (2022). For RGB images from the ImageNet and HAM10000 datasets, we set $\epsilon = 8/255$, whereas for greyscale images from the Br35h and COVID-QU-Ex datasets, we set $\epsilon = 6/225$ Dong et al. (2025). Consistent with prior studies, we set $K = 5000$ Williams & Li (2023a;b) for all attacks. As outlined in Section 3, our approach involves three adjustable parameters: $\sigma$, $N$, and 'Max Diameter'. Here, $\sigma$ governs the exploration of the method, $N$ signifies the number of shapes used to create the perturbation, and 'Max Diameter' determines the largest possible size of the perturbation circles. The specific values for these parameters are listed in Table 4, with justification provided in Section 4.5.

**Performance Metrics:** For evaluating the effectiveness of the proposed method, we follow previous works and adopt $PCC$ for measuring of the distortion caused to the attribution maps Huang et al. (2023). $PCC$ values near 1 indicate strong positive correlation, values close to 0 imply no correlation, and values approaching $-1$ suggest strong negative correlation. Additionally, we report the percentage of generated adversarial images that meet the constraints specified in (1) and (2). To ensure fair comparison, distortion measurements are only conducted on images that satisfy their respective task constraints. For ranking the robustness of XAI methods, we utilize the dominance relations described in Definitions 1 and 2 to evaluate and rank the considered XAI methods.

Given the stochastic nature of our method, each experiment is repeated over 10 different random seeds. For each metric, we aggregate its value across all model architectures. To statistically verify whether the results achieved by our method are significantly different to other algorithms, we employ the Wilcoxon signed-rank test Wilcoxon (1992) at a 5% significance level, as is standard practice within the evolutionary optimization field Williams & Li (2023a); Storn & Price (1997); Deb (2001).

## 4.2 Results Analysis

To ensure a fair comparison among EvoAttack, SAFARI, and NES, we fix the population size of SAFARI and NES to 50, which corresponds to 100 iterations of their optimization cycles. For Square Attack, we adapt its selection strategy by replacing it with the domination functions defined in 1 and 2.

**Constraint satisfaction:** Table 1 reports statistical results for the proposed method and comparative attacks; arrows indicate whether larger or smaller metric values are preferable. Task 1 enforces image misclassification, whereas Task 2 requires the original correct classification to be preserved. The Task 2 satisfaction rates below 100% for EvoAttack and SAFARI indicate that even random perturbations sometimes

Table 1: Table presents the Pearson Correlation Coefficient (PCC) along with the percentage of images that satisfy the respective task constraint when attacking images from the HAM10000, Br35h and COVID-QU-Ex datasets. We provide the mean each metric over 10 runs and its variance inside the brackets.

| DeepLIFT | Task 1 | | Task 2 | |
|---|---|---|---|---|
| Method | Constraint Satisfied ($\uparrow$) | PCC ($\uparrow$) | Constraint Satisfied ($\uparrow$) | PCC ($\downarrow$) |
| Square-Attack | **84.87%(2.054)** | $0.52(0.1)^{\ddagger}$ | 77.34%(2.046) | $0.29(0.097)^{\ddagger}$ |
| EvoAttack | 83.91%(1.718) | $\mathbf{0.7(0.08)}^{\dagger}$ | **78.23%(2.164)** | $\mathbf{-0.03(0.08)}^{\dagger}$ |
| NES | $30.84\%(1.53)^{\ddagger}$ | $0.2(0.096)^{\ddagger}$ | $61.02\%(2.363)^{\ddagger}$ | $0.35(0.105)^{\ddagger}$ |
| SAFARI | $36.81\%(1.639)^{\ddagger}$ | $0.46(0.117)^{\ddagger}$ | $65.72\%(1.783)^{\ddagger}$ | $0.53(0.114)^{\ddagger}$ |
| Saliency | Task 1 | | Task 2 | |
| Method | Constraint Satisfied ($\uparrow$) | PCC ($\uparrow$) | Constraint Satisfied ($\uparrow$) | PCC ($\downarrow$) |
| Square-Attack | **83.07%(1.979)** | $0.4(0.097)^{\ddagger}$ | 81.46%(1.631) | $0.24(0.097)^{\ddagger}$ |
| EvoAttack | 81.94%(2.463) | $\mathbf{0.57(0.077)}^{\dagger}$ | **83.05%(1.489)** | $\mathbf{0.0(0.081)}^{\dagger}$ |
| NES | $30.85\%(1.982)^{\ddagger}$ | $0.2(0.092)^{\ddagger}$ | $61.28\%(1.392)^{\ddagger}$ | $0.29(0.112)^{\ddagger}$ |
| SAFARI | $32.25\%(2.114)^{\ddagger}$ | $0.35(0.088)^{\ddagger}$ | $64.91\%(2.328)^{\ddagger}$ | $0.41(0.099)^{\ddagger}$ |
| Grad-CAM | Task 1 | | Task 2 | |
| Method | Constraint Satisfied ($\uparrow$) | PCC ($\uparrow$) | Constraint Satisfied ($\uparrow$) | PCC ($\downarrow$) |
| Square-Attack | **82.71%(2.022)** | $0.36(0.149)^{\ddagger}$ | 86.5%(1.949) | $0.3(0.161)^{\ddagger}$ |
| EvoAttack | 81.79%(2.063) | $\mathbf{0.88(0.136)}^{\dagger}$ | **87.62%(2.186)** | $\mathbf{-0.57(0.195)}^{\dagger}$ |
| NES | $31.08\%(1.802)^{\ddagger}$ | $0.18(0.269)^{\ddagger}$ | $59.95\%(2.465)^{\ddagger}$ | $0.43(0.211)^{\ddagger}$ |
| SAFARI | $33.6\%(2.034)^{\ddagger}$ | $0.31(0.256)^{\ddagger}$ | $62.0\%(1.346)^{\ddagger}$ | $0.55(0.214)^{\ddagger}$ |
| Shapley | Task 1 | | Task 2 | |
| Method | Constraint Satisfied ($\uparrow$) | PCC ($\uparrow$) | Constraint Satisfied ($\uparrow$) | PCC ($\downarrow$) |
| Square-Attack | **82.69%(2.229)** | $0.52(0.12)^{\ddagger}$ | 75.06%(1.82) | $0.29(0.11)^{\ddagger}$ |
| EvoAttack | 81.67%(1.777) | $\mathbf{0.73(0.092)}^{\dagger}$ | **75.83%(1.767)** | $\mathbf{0.04(0.107)}^{\dagger}$ |
| NES | $30.58\%(2.39)^{\ddagger}$ | $0.25(0.129)^{\ddagger}$ | $61.3\%(1.974)^{\ddagger}$ | $0.39(0.125)^{\ddagger}$ |
| SAFARI | $32.98\%(2.564)^{\ddagger}$ | $0.46(0.112)^{\ddagger}$ | $63.05\%(1.678)^{\ddagger}$ | $0.56(0.108)^{\ddagger}$ |
| Input x Gradient | Task 1 | | Task 2 | |
| Method | Constraint Satisfied ($\uparrow$) | PCC ($\uparrow$) | Constraint Satisfied ($\uparrow$) | PCC ($\downarrow$) |
| Square-Attack | **84.17%(1.906)** | $0.4(0.105)^{\ddagger}$ | 75.11%(1.963) | $0.24(0.103)^{\ddagger}$ |
| EvoAttack | 82.92%(1.95) | $\mathbf{0.57(0.072)}^{\dagger}$ | **76.24%(1.759)** | $\mathbf{0.05(0.08)}^{\dagger}$ |
| NES | $30.57\%(1.302)^{\ddagger}$ | $0.15(0.084)^{\ddagger}$ | $60.33\%(1.569)^{\ddagger}$ | $0.28(0.107)^{\ddagger}$ |
| SAFARI | $36.97\%(2.364)^{\ddagger}$ | $0.35(0.099)^{\ddagger}$ | $64.2\%(1.301)^{\ddagger}$ | $0.41(0.104)^{\ddagger}$ |
| Grad-CAM++ | Task 1 | | Task 2 | |
| Method | Constraint Satisfied ($\uparrow$) | PCC ($\uparrow$) | Constraint Satisfied ($\uparrow$) | PCC ($\downarrow$) |
| Square-Attack | **83.71%(1.672)** | $0.66(0.144)^{\ddagger}$ | 83.69%(2.299) | $0.33(0.168)^{\ddagger}$ |
| EvoAttack | 81.72%(2.135) | $\mathbf{0.89(0.065)}^{\dagger}$ | **84.73%(1.664)** | $\mathbf{-0.46(0.243)}^{\dagger}$ |
| NES | $30.37\%(2.115)^{\ddagger}$ | $0.28(0.243)^{\ddagger}$ | $60.52\%(2.197)^{\ddagger}$ | $0.44(0.21)^{\ddagger}$ |
| SAFARI | $35.81\%(1.849)^{\ddagger}$ | $0.61(0.144)^{\ddagger}$ | $65.58\%(2.03)^{\ddagger}$ | $0.6(0.155)^{\ddagger}$ |
| LIME | Task 1 | | Task 2 | |
| Method | Constraint Satisfied ($\uparrow$) | PCC ($\uparrow$) | Constraint Satisfied ($\uparrow$) | PCC ($\downarrow$) |
| Square-Attack | **86.06%(4.143)** | $0.33(0.14)^{\ddagger}$ | 88.88%(3.893) | $0.27(0.1)^{\ddagger}$ |
| EvoAttack | 85.14%(5.216) | $\mathbf{0.88(0.071)}^{\dagger}$ | **90.4%(5.177)** | $\mathbf{-0.6(0.091)}^{\dagger}$ |
| NES | $33.91\%(3.96)^{\ddagger}$ | $0.21(0.13)^{\ddagger}$ | $69.38\%(3.623)^{\ddagger}$ | $0.41(0.103)^{\ddagger}$ |
| SAFARI | $38.23\%(4.944)^{\ddagger}$ | $0.28(0.138)^{\ddagger}$ | $50.19\%(4.238)^{\ddagger}$ | $0.55(0.09)^{\ddagger}$ |

$^{\dagger}$ denotes the performance of the method significantly outperforms the compared methods according to the Wilcoxon signed-rank test Wilcoxon (1992) at the 5% significance level; $^{\ddagger}$ denotes the corresponding method is significantly outperformed by the best performing method (shaded).

cause the DNN to misclassify, preventing those attacks from producing perturbations that meet the Task 2 constraint. For NES, the ability to satisfy either constraint additionally depends on the relative weighting used to combine the classification and attribution objectives.

Overall, EvoAttack outperforms both SAFARI Huang et al. (2023) and NES Tamam et al. (2023) across XAI methods and datasets, particularly under tight query budgets. As noted in Section 1, many existing approaches implicitly assume access to large query budgets, and their effectiveness degrades substantially when queries are limited. In our experiments, NES and SAFARI frequently fail to induce adversarial misclassification (Task 1) across datasets, consistent with prior observations about the difficulty of generating adversarial

| XAI Method | HAM10000 | | Br35h | | COVID-QU-Ex | |
|---|---|---|---|---|---|---|
| | Task 1 | Task 2 | Task 1 | Task 2 | Task 1 | Task 2 |
| DeepLIFT | 4.19 (0.982) | 4.15 (1.269) | 4.02 (1.535) | 4.9 (1.368) | 4.04 (1.043) | **2.71(0.825)** |
| Grad-CAM | 5.48 (1.095) | 5.88 (1.649) | 4.14 (1.529) | 5.84 (1.839) | 6.2 (0.965) | 5.71 (1.285) |
| SHAPLEY | 4.87 (0.875) | 3.12 (0.968) | 4.69 (1.467) | 5.0 (1.535) | 4.31 (1.012) | 3.04 (0.943) |
| Input x Gradient | **2.43(0.86)** | **2.82(0.948)** | **3.93(1.437)** | **4.15(1.637)** | 3.34 (0.947) | 2.9 (1.02) |
| Grad-CAM++ | 6.17 (1.516) | 5.49 (1.669) | 4.42 (1.373) | 5.16 (2.013) | 6.52 (0.726) | 5.89 (1.489) |
| Saliency | 3.38 (0.963) | 2.93 (1.037) | 4.51 (1.34) | 4.95 (1.367) | **2.54(0.79)** | 4.35 (1.262) |

Table 2: Table presents the robustness ranking of XAI methods across each task and dataset using the proposed EvoAttack as the attack method. We provide the mean and variance of each metric over 10 runs.

images under constrained-query settings. By contrast, EvoAttack achieves substantially higher success rates in producing adversarial examples while better respecting the query and classification constraints.

Square Attack shares conceptual similarities with EvoAttack in that it also employs shape-based perturbations. Our experimental results indicate that Square Attack achieves a comparable success rate in inducing misclassification. This effectiveness can be attributed to two main factors: (i) its perturbations are bounded within the range $[-\epsilon, \epsilon]$, a constraint known to be effective for generating adversarial examples, and (ii) its square-shaped perturbations have been shown to be particularly effective in misleading classifiers.

When evaluating performance under the Task 2 condition, the differences among NES, SAFARI, and EvoAttack are less pronounced. Since Task 2 requires perturbations that preserve the original classification, this outcome can be attributed to the inherent robustness of the underlying DNN classifiers to small pixel-level perturbations. Compared with Square Attack, EvoAttack achieves higher success under this constraint: although Square Attack is highly effective at inducing misclassification, the strength of its perturbations diminishes its ability to maintain correct classifications.

When analyzing constraint satisfaction rates across datasets (see Tables 11, 13, 12, and 14 in the Appendix), all attack methods exhibit reduced performance on the Br35h dataset. Despite the presence of clearly defined regions of interest across all datasets, diagnostic cues in Br35h images tend to be more explicit—such as the visible presence of a tumour. Consequently, the classifiers may have learned more robust features (e.g., tumour presence) rather than relying on non-robust or spurious correlations, a phenomenon commonly referred to as shortcut learning Wang et al. (2024).

Finally, only minor discrepancies in constraint satisfaction rates are observed across different XAI methods. This uniformity is expected for EvoAttack, Square Attack, and SAFARI, as all prioritize satisfying the task constraints before optimizing for explanation distortion.

**Explanation distortion:** In assessing the distortions to explanations caused by the attack methods, EvoAttack consistently outperforms Square-Attack, SAFARI and NES across all experimental scenarios, demonstrating significant superiority in the majority of cases. Notably, there is a pronounced performance disparity when targeting XAI methods that generate less granular, more region-focused explanations, such as Grad-CAM, Grad-CAM++, and LIME. Conversely, when attacking more granular XAI methods like DeepLIFT, Saliency maps, SHAPLEY, and Input x Gradient, the performance gap between EvoAttack and the other attack methods narrows. This outcome is anticipated due to the less granular perturbation strategy employed by EvoAttack, emphasizing the importance of accounting for the granularity of XAI explanations when designing perturbations. However, it also underscores the adaptability of the proposed method to effectively target both granular and non-granular XAI methods.

Despite its success in satisfying the task constraints, Square Attack struggled to distort the attribution maps produced by the XAI methods. This limitation can be attributed to several factors. First, Square Attack samples perturbations exclusively on the constraint boundary $[-\epsilon, +\epsilon]$, which is effective for classification attacks but can be overly restrictive for more complex objectives such as explanation distortion. Second, the attack proceeds by initially perturbing the entire image with vertical strips and then adding one square at a time; however, once a square is accepted, it cannot be removed, making the optimization process being prone to local optima. In contrast, EvoAttack can flexibly adjust perturbations to cover either broad regions

or localized areas of the image. Third, as the attack progresses, Square Attack iteratively decreases the size of its perturbation squares. While this strategy is advantageous for inducing misclassification, it proves less effective for explanation methods that emphasize coarse-grained features (e.g., Grad-CAM), where fine perturbations exert minimal influence. For more fine-grained XAI methods, the sparse structure of Square Attack's perturbations similarly reduces effectiveness. EvoAttack, however, adapts its perturbations to varying levels of granularity through its attack parameters, enabling stronger distortion of attribution maps.

Comparing task performances, the attack methods generally achieve greater success in maximizing the distortion of explanations for correctly classified images rather than minimizing distortion for misclassified ones. This indicates that XAI-generated explanations are more susceptible to Task 2 attacks, where the classification remains accurate, but the explanation is distorted. Although this scenario poses less risk to patient safety due to correct disease classification, it could undermine trust in AI systems among medical practitioners, potentially reducing their willingness to rely on AI for assistance Rosenbacke et al. (2024a).

Additionally, performances vary across different XAI methods among the attack strategies. EvoAttack notably manipulates less granular explanations, such as Grad-CAM, Grad-CAM++, and LIME, more effectively in all experimental setups compared to the granular methods, underscoring their specific vulnerabilities to structured perturbations. Among granular XAI methods, the greatest challenge appears in attacking Saliency maps and Input x Gradient, particularly for Task-1. For example, when targeting images from the HAM10000 and COVID-QU-Ex datasets, no attack achieves an average $PCC$ value above 0.6, which is a benchmark for consistency between explanations. Conversely, for Task 2, EvoAttack is able to reduce the average $PCC$ of those XAI values below 0.4, indicating inconsistent explanations Huang et al. (2023), highlighting the greater vulnerability of XAI methods when distorting correctly classified images.

More visual comparisons of adversarial images and explanations is provided in Section A.6 of the Appendix. We also compare the performance of the proposed and compared attack methods over varying query budget $K$ in Section A.8 along with their attack speed in Section A.10.

### 4.3   XAI Robustness Comparison

The findings in Section 4.2 highlight the superior performance of the proposed method over existing attack techniques, showcasing its utility for robustness evaluations. However, the combination of the task's objective and constraint (as detailed in Section 3) makes it difficult to use a single metric value for ranking. Therefore, to rank the robustness of the targeted XAI methods, we employ the task-specific domination relations defined in 1 and 2.

For each attacked image, we utilize non-dominated sorting Deb et al. (2002) to rank each XAI method based on the performance of EvoAttack in targeting the image, where lower ranks correspond to poorer attack results, indicating greater robustness of the XAI method. This procedure is repeated across all 100 images for each model architecture. We repeat this procedure over the 10 different random seeds, with the average rank serving as a measure of the XAI method's overall performance.

The final robustness ranking of XAI methods, as presented in Table 2, indicates that Input x Gradient frequently achieves better average rankings across most attack instances. Despite the Saliency method attaining better average PCC values, it achieves higher overall rankings on 2 out of 3 datasets. This discrepancy arises because the metric values in Section 4.2 focused solely on attack instances that satisfied the constraints, whereas this ranking accounts for all evaluated images. For example, during Task-1 attacks on images from the HAM10000 dataset (see Table 12 within the Appendix), the Saliency method shows greater resistance to attacks, resulting in a higher average $PCC$ value. By employing the domination relation, the ranking considers the attack's ability to satisfy the constraint, demonstrating that EvoAttack achieves lower constraint satisfaction when targeting Input x Gradient compared to Saliency, leading to Input x Gradient's superior average rank. Finally, in Task-1 attacks on COVID-QU-Ex images, the Saliency method demonstrates superior robustness, while DeepLIFT proves most resilient against Task-2 attacks.

Table 3: Table presents the Pearson Correlation Coefficient (PCC) along with the percentage of images that satisfy the respective task constraint when attacking XAI method applied to adversarial trained classifiers. We provide the mean and variance of each metric over 10 runs.

| DeepLIFT | Task 1 | | Task 2 | |
|---|---|---|---|---|
| Dataset | Constraint Satisfied (↑) | PCC (↑) | Constraint Satisfied (↑) | PCC (↓) |
| HAM10000 | 98.06 (2.04) | 0.71 (0.118) | 97.01 (2.544) | 0.08 (0.162) |
| Br35h | 56.00 (2.376) | 0.66 (0.148) | 56.99 (2.182) | 0.56 (0.158) |
| COVID-QU-Ex | 98.01 (2.577) | 0.57 (0.154) | 61.08 (2.508) | -0.0 (0.235) |
| Saliency | Task 1 | | Task 2 | |
| Dataset | Constraint Satisfied (↑) | PCC (↑) | Constraint Satisfied (↑) | PCC (↓) |
| HAM10000 | 98.07 (2.214) | 0.54 (0.112) | 98.91 (2.782) | 0.17 (0.16) |
| Br35h | 59.29 (2.816) | 0.65 (0.124) | 59.0 (2.218) | 0.51 (0.146) |
| COVID-QU-Ex | 98.95 (2.46) | 0.46 (0.11) | 67.01 (2.86) | -0.1 (0.083) |
| Grad-CAM | Task 1 | | Task 2 | |
| Dataset | Constraint Satisfied (↑) | PCC (↑) | Constraint Satisfied (↑) | PCC (↓) |
| HAM10000 | 98.07 (2.034) | 0.77 (0.238) | 99.94 (2.531) | -0.36 (0.391) |
| Br35h | 40.0 (2.46) | 0.60 (0.209) | 48.0 (2.334) | 0.68 (0.365) |
| COVID-QU-Ex | 98.92 (2.31) | 0.78 (0.268) | 75.06 (2.367) | -0.73 (0.481) |
| Shapley | Task 1 | | Task 2 | |
| Dataset | Constraint Satisfied (↑) | PCC (↑) | Constraint Satisfied (↑) | PCC (↓) |
| HAM10000 | 97.9 (2.185) | 0.76 (0.11) | 96.91 (2.139) | 0.13 (0.176) |
| Br35h | 41.0 (2.687) | 0.63 (0.154) | 42.0 (2.385) | 0.56 (0.139) |
| COVID-QU-Ex | 98.96 (2.866) | 0.59 (0.151) | 60.93 (2.174) | -0.0 (0.223) |
| Input x Gradient | Task 1 | | Task 2 | |
| Dataset | Constraint Satisfied (↑) | PCC (↑) | Constraint Satisfied (↑) | PCC (↓) |
| HAM10000 | 97.94 (2.476) | 0.48 (0.123) | 96.99 (2.005) | 0.12 (0.147) |
| Br35h | 52.03 (2.82) | 0.67 (0.131) | 41.0 (2.913) | 0.56 (0.118) |
| COVID-QU-Ex | 97.95 (2.708) | 0.52 (0.1) | 60.9 (2.6) | 0.01 (0.087) |
| Grad-CAM++ | Task 1 | | Task 2 | |
| Dataset | Constraint Satisfied (↑) | PCC (↑) | Constraint Satisfied (↑) | PCC (↓) |
| HAM10000 | 98.02 (2.541) | 0.86 (0.154) | 99.9 (2.416) | -0.33 (0.401) |
| Br35h | 46.0 (2.105) | 0.49 (0.259) | 46.0 (2.344) | 0.69 (0.34) |
| COVID-QU-Ex | 97.92 (2.327) | 0.87 (0.154) | 75.06 (2.933) | -0.73 (0.482) |
| LIME | Task 1 | | Task 2 | |
| Dataset | Constraint Satisfied (↑) | PCC (↑) | Constraint Satisfied (↑) | PCC (↓) |
| HAM10000 | 99.31 (1.409) | 0.77 (0.101) | 99.4 (0.18) | -0.41 (0.392) |
| Br35h | 38.11 (1.201) | 0.49 (0.259) | 46.0 (2.344) | 0.69 (0.34) |
| COVID-QU-Ex | 97.92 (2.327) | 0.87 (0.154) | 75.06 (2.933) | -0.73 (0.482) |

## 4.4 Evaluation of Adversarial Training

The results in Section 4.2 demonstrate the effectiveness of the proposed attack in compromising XAI methods and outperforming existing attacks. Since adversarial training has emerged as a promising defence against adversarial attacks, we next evaluate its potential to improve the robustness of XAI methods against EvoAttack. A comparison with the black-box defence mechanisms proposed by Qin et al. and Cohen et al. is further provided in Section A.9 of the appendix.

**Adversarial Training Setup:** We implement adversarial training on the HAM10000, Br35h, and COVID-QU-Ex datasets following a similar procedure established in prior work Madry et al. (2017); Chernyak et al. (2021). In each training iteration, all images are augmented with random EvoAttack perturbations, producing batches that contain both benign and adversarial examples and thereby doubling the effective batch size. To expose the DNN to a diverse range of perturbation structures, the parameters $N$ and Maximum Diameter are randomly sampled for each perturbation from the grid used in our ablation study. A detailed description of the adversarial training procedure, along with the performance metrics of the adversarially trained classifiers, is provided in Section A.4.

**Task 1:** The results presented in Table 3 illustrates the impact of adversarial training on different XAI methods. First, we see that adversarially trained HAM10000 DNN classifiers become more vulnerable to

adversarial attacks. This phenomenon is likely attributed to the large class imbalance in the HAM10000 dataset, which has shown to be an issue for adversarial training Wang et al. (2022). Comparing with Br35h and COVID-QU-Ex classifiers, we see adversarial training has improved their robustness against the proposed EvoAttack.

Comparing the performance of the different XAI methods, we observe that adversarial training consistently improves robustness across all methods. The effect is particularly pronounced for the less granular methods, such as Grad-CAM, Grad-CAM++, and LIME, which exhibit larger gains in robustness. Nevertheless, these methods remain among the least vulnerable overall. For the more granular XAI methods, adversarial training substantially enhances the robustness of Saliency Maps and Input $\times$ Gradient, reducing their $PCC$ values to around or below 0.6 across all three datasets. This suggests that, while adversarial explanations are not entirely inconsistent with their benign counterparts, their similarity falls close to or below the threshold of consistency, implying that distortions in the attribution maps may become perceptible.

**Task 2:** Similar to Task 1, we witness an enhancement in robustness across all XAI methods which can be described by the increased average $PCC$ values. In the case of Br35h images, EvoAttack's ability to meet the constraint is diminished, reflecting improved robustness across XAI methods, with all average $PCC$ values exceeding 0.5. This suggests that EvoAttack faces challenges in altering the original explanation to be inconsistent while ensuring the classifier predicts accurately.

Similar observations arise with the COVID-QU-Ex dataset, where EvoAttack's success in meeting the constraint is reduced, while the average $PCC$ values for most XAI methods increase. This indicates that adversarial training has degraded EvoAttack's ability to distort explanations while retaining accurate classifier predictions.

Despite the impact of adversarial training in improving the robustness across XAI methods, EvoAttack is still able to distort attribution maps to $PCC$ values of below 0.4, which indicates that inconsistency was achieved.

### 4.5 Ablation Study

The proposed method incorporates three tunable parameters: $N$, $\sigma$ and the maximum circle diameter 'Max Diameter' expressed as a percentage of the image. We employ a grid search over the parameter space to determine their optimal values. Specifically, we explore $N \in 100, 300$, $\sigma \in 0.1, 0.2, 0.3$, and maximum circle diameters with ranges $20\%, 30\%, 40\%, 50\%$ of the original image size. These parameter ranges are based on commonly set values used in the evolutionary Skiscim & Golden (1983) and computational art Tian & Ha (2022) fields. To evaluate the performance of each parameter configuration, we conduct attacks on each explanation for each task using a VGG-16 ImageNet classifier with 100 correctly classified images from the validation set. To compare the performance of different parameter configurations, we employ the same methodology as described in Section 4.3.

**Configuration Performance Analysis:** As illustrated in Table 4, the optimal configuration varies significantly across XAI methods and tasks. Similar to previous studies that highlight the lack of consensus among XAI methods, this suggests that the vulnerabilities between them may differ. Nonetheless, some patterns emerge across different XAI methods. Firstly, for Task 1, which aims to induce misclassification while preserving the explanation, the best-performing configurations consistently feature a maximum circle diameter of 80% of the image size. Conversely, configurations with smaller circle diameters perform worse. This indicates that utilizing larger local perturbations (i.e., larger circles) is more effective in influencing the DNN classifier while minimizing distortion in the attribution map.

Another observed pattern is that all optimal configurations use $N = 100$ circles. A likely explanation for this is that adding more shapes increases the number of variables involved, which might demand a larger computational budget for effective optimization Williams et al. (2021); Eltaeib & Mahmood (2018). Keeping a constant budget might lead to prematurely halting the optimization process, thus affecting performance negatively.

For Task 2, which aims to distort the explanation while maintaining correct classification, we observe greater variation in the best performing configurations. However, similar configuration performances are seen across

|  | Task 1 | | | Task 2 | | |
| --- | --- | --- | --- | --- | --- | --- |
| XAI Method | N | Max Diameter (%) | $\sigma$ | N | Max Diameter (%) | $\sigma$ |
| DeepLIFT | 100 | 80.0 | 0.1 | 300 | 60.0 | 0.3 |
| Saliency | 100 | 80.0 | 0.3 | 200 | 70.0 | 0.2 |
| Grad-CAM | 100 | 80.0 | 0.3 | 100 | 80.0 | 0.2 |
| LIME | 100 | 80.0 | 0.3 | 100 | 80.0 | 0.2 |
| Shapley | 100 | 80.0 | 0.3 | 300 | 60.0 | 0.3 |
| Input x Gradient | 100 | 80.0 | 0.1 | 300 | 60.0 | 0.3 |
| Grad-CAM++ | 100 | 80.0 | 0.1 | 100 | 80.0 | 0.3 |

Table 4: Chosen EvoAttack parameters for attacking the considered XAI methods.

different XAI methods. Specifically, the results indicate that XAI methods that produce similar granularity in attribution maps are effectively attacked with comparable parameter setups. For example, when targeting highly granular XAI methods of DeepLIFT, Input x Gradient, and SHAPLEY (see Figure 2), the proposed method achieves superior performance using smaller diameter circles. Conversely, XAI methods Grad-CAM, Grad-CAM++, and SHAPLEY, which highlight broader regions rather than individual pixels, are more susceptible to larger circular perturbations. This behaviour stems from altering the perturbations' granularity by changing circle size, with smaller circles constructing perturbations more closely resembling pixel-level perturbations.

A surprising result from the ablation study was the performance of parameter configurations when targeting the Saliency XAI method. Unlike other granular methods, the Saliency method proved more vulnerable to medium to large circle perturbations, while remaining robust against smaller circles. This difference might be attributed to its level of granularity. Whereas SHAPLEY and DeepLIFT produce sparse maps emphasizing specific pixels, Saliency maps highlight broader regions. Although the Input x Gradient method also emphasizes broader areas, its multiplication with the input image may also have an impact, resulting in being vulnerable to perturbations constructed from smaller circles.

These results underscore the advantages of the proposed attack method. By adjusting the size of the circular shapes, EvoAttack effectively manages the trade-off related to XAI granularity when targeting explanation methods—an aspect that existing strategies lack. In conclusion, we recommend adopting the optimal configurations detailed in Table 4 for the XAI methods considered in this study. We provide the performance across all configurations in Figure 7 within the Appendix.

## 5 Conclusion, Limitations, and Future Work

**Conclusion:** This paper introduces a novel adversarial attack designed to attack computer vision XAI methods. Unlike existing approaches that modify each pixel of the benign image, our method constructs adversarial perturbations by concatenating RGB-valued circular shapes. We optimize the parameters of these shapes using a (1+1)-evolutionary strategy, a widely used optimization heuristic in evolutionary computation. To enhance the attack's efficacy, we conducted an ablation study assessing the influence of various parameters on EvoAttack's performance across several XAI methods. The results demonstrate that larger circles effectively manipulate less granular XAI methods like Grad-CAM, Grad-CAM++, and LIME, whereas smaller circles yield better results against more granular XAI methods such as DeepLIFT and Input x Gradient. Compared to state-of-the-art attacks, the proposed method consistently outperforms them in distorting XAI attributions maps across all attack setups, showcasing its efficiency and effectiveness.

Then, we leverage the EvoAttack method to evaluate and rank the robustness of XAI methods. Given the complexity of assessing XAI methods using both PCC and constraint metrics, we employed the EvoAttack domination relation for each task to rank the resistance of XAI methods with respect to each attacked image. By averaging the rank of each XAI method across all attacked images and underlying classifiers, we formulate a comprehensive ranking. This approach allowed us to incorporate all data regarding XAI distortion alongside constraint satisfaction, culminating in a unified ranking table for each dataset.

To counter the proposed attack, we developed an adversarial training procedure that incorporates random EvoAttack-style perturbations into the training process. Attacking adversarially trained models revealed enhanced robustness in most XAI methods; however, we noted a decrease in classifier robustness on the

HAM10000 dataset, accompanied by improved XAI robustness. In the Task 2 attack scenario, adversarially trained classifiers decreased EvoAttack's effectiveness in redirecting the DNN classifier towards correct predictions, suggesting that while the classifier became more susceptible to random EvoAttack perturbations, it also developed a stronger resistance to subsequent manipulations. This study underscores the potential of adversarial training as a crucial strategy for defending against attacks, in addition to the utility of considering both classifier and XAI performance when evaluating the robustness of human-in-the-loop systems. Nevertheless, our study highlights the need for further exploration, marking this as a vital area for future research.

**Limitations and Future Work:** This study focuses on evaluating and enhancing the robustness of XAI techniques, with an emphasis on medical imaging datasets. While the proposed method demonstrates promising results, there are limitations and avenues for future research. Firstly, the hyper-parameters were optimized using a basic grid-search approach, aimed at analysing the impact of varying parameter values. In future work, more advanced hyper-parameter optimization frameworks, such as Bayesian Optimization or methods assisted by large-language models, should be explored Snoek et al. (2012); Zhang et al. (2023).

Additionally, this study employed a predefined $l_\infty$ constraint $\epsilon$ based on existing recommendations. Exploring minimum-norm attacks, which identify the minimal $\epsilon$ value necessary to compromise an AI system could provide deeper insights into the XAI and classifier vulnerabilities Williams & Li (2023a). Moreover, while this research examined perturbations that modify all pixels, future studies should consider alternative perturbations, such as sparse attacks, where only a limited number of pixels are altered.

Finally, this study assumes access to both classifier output probabilities and attribution maps. In practice, such access may be restricted. Evaluating robustness under these conditions could be guided by decision-only or transfer-based attack paradigms Ilyas et al. (2018); Papernot et al. (2016); Guo et al. (2019); Cheng et al. (2019), enabling the development of novel robustness assessments tailored to explainable AI systems.

Our experiments highlight the potential of adversarial training in mitigating the impact of EvoAttack on both classifier and XAI robustness. At the same time, they reveal EvoAttack's ability to manipulate explanations to appear either consistent or inconsistent, depending on the task, while still satisfying the respective constraints. Future work should therefore explore more sophisticated adversarial training strategies aimed at strengthening the robustness of both DNN classifiers and XAI methods. One promising direction is to replace random perturbations with targeted EvoAttack perturbations during adversarial training, even under limited query budgets, to expose models to more harmful perturbations and improve resilience.

To encourage further research in this domain, we will publicly release our implementation, datasets, and evaluation scripts upon acceptance of the paper.

## Broader Impact Statement

In this work, we introduce a novel adversarial attack method against explainable AI (XAI) techniques that can account of the varying granularities in explanations, as well as reducing the dimension of the search space.

We apply our proposed attack to the robustness ranking of various XAI methods across three different medical image datasets. This study underscores the necessity of evaluating both classifier and XAI system robustness.

Within the healthcare domain, our research demonstrates two significant risks: first, the possibility of medical professionals trusting incorrect AI diagnoses due to seemingly plausible explanations, potentially endangering patient safety; and second, the risk of decreasing trust between healthcare practitioners and AI systems due to distorted explanations of accurate diagnoses, which could slow down the diagnostic process by necessitating additional human evaluations. To address these risks, we explore different mitigation strategies, such as adversarial training, which showed promise in enhancing the resilience of XAI systems against adversarial threats.

We hope our work will lead to further research into adversarial training strategies and encourage practitioners to rigorously test the robustness of XAI systems before deployment. Our ultimate goal is to advance the safe and effective integration of AI in critical domains like healthcare.

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
