

Figure 6: Flowchart demonstrates the process of the proposed EvoAttack method. The proposed method iteratively generates a single child solution by applying random changes to its parent. If the parent solution is dominated (defined in Section 3.4) by its child, it is replaced for the next generation, otherwise the parent remains.

# A Appendix

## A.1 Proposed Method

## A.2 DNN Training

In this section we report the final performance metrics of the trained DNN classifiers. As outlined in Section 4 we fine-tune each chosen ImageNet pre-trained model for 10 epochs, using a batch-size of 32, and a learning-rate of $1e - 4$ using an ADAM optimiser Kingma & Ba (2015) and a Cross Entropy Loss.

**ImageNet:** In this work we make use of ImageNet pre-trained DNN classifiers available from the pytorch library Paszke et al. (2019) (for standard implementation) and the authors original implementation of Böhle et al. (2024). We refer the reader to https://pytorch.org/vision/stable/models.html and https://github.com/B-cos/B-cos-v2 for the performance metrics of used pre-trained models.

**HAM10000:** This dataset consists of 10000 images of common pigmented skin lesions across 7 classes of difference cancerous lesions. We observe that the fine-tuned model achieve test-set performance metrics similar to existing methods within the literature Nadipineni (2020). We outline the performance metrics of our trained models in Table 5.

**COVID-QU-Ex:** This dataset contains a total of $33,920$ chest X-rays covering 3 classes of COVID-19, Pneumonia and Normal. We outline the performance metrics of our trained models in Table 6.

**Br35h:** This dataset contains a total of 3000 images of brain MRI scans with and without a tumor. We present the performance metrics of the trained models in Table 7.

| AlexNet | | | | |
|---|---|---|---|---|
| Class | Precision | Recall | f1-score | Support |
| 0 | 0.66 | 0.71 | 0.68 | 65 |
| 1 | 0.79 | 0.77 | 0.78 | 109 |
| 2 | 0.66 | 0.73 | 0.69 | 215 |
| 3 | 0.44 | 0.72 | 0.55 | 25 |
| 4 | 0.50 | 0.78 | 0.60 | 206 |
| 5 | 0.96 | 0.85 | 0.90 | 1354 |
| 6 | 0.87 | 0.90 | 0.88 | 29 |

Accuracy: 0.82

| MobileNet | | | | |
|---|---|---|---|---|
| Class | Precision | Recall | f1-score | Support |
| 0 | 0.72 | 0.60 | 0.66 | 65 |
| 1 | 0.78 | 0.85 | 0.81 | 109 |
| 2 | 0.77 | 0.71 | 0.74 | 215 |
| 3 | 0.85 | 0.68 | 0.76 | 25 |
| 4 | 0.62 | 0.73 | 0.67 | 206 |
| 5 | 0.94 | 0.93 | 0.93 | 1354 |
| 6 | 0.96 | 0.93 | 0.95 | 29 |

Accuracy: 0.87

| VGG-16 | | | | |
|---|---|---|---|---|
| Class | Precision | Recall | f1-score | Support |
| 0 | 0.74 | 0.69 | 0.71 | 65 |
| 1 | 0.76 | 0.82 | 0.79 | 109 |
| 2 | 0.67 | 0.79 | 0.72 | 215 |
| 3 | 0.88 | 0.56 | 0.68 | 25 |
| 4 | 0.48 | 0.81 | 0.60 | 206 |
| 5 | 0.96 | 0.84 | 0.90 | 1354 |
| 6 | 0.90 | 0.97 | 0.93 | 29 |

Accuracy: 0.82

Table 5: Test-set performance metrics of pre-trained ImageNet models fine-tuned on the HAM1000 dataset.

| AlexNet | | | | |
|---|---|---|---|---|
| Class | Precision | Recall | f1-score | Support |
| 0 | 0.96 | 0.91 | 0.94 | 2395 |
| 1 | 0.89 | 0.94 | 0.91 | 2253 |
| 2 | 0.90 | 0.89 | 0.89 | 2140 |

Accuracy: 0.92

| MobileNet | | | | |
|---|---|---|---|---|
| Class | Precision | Recall | f1-score | Support |
| 0 | 0.96 | 0.97 | 0.97 | 2395 |
| 1 | 0.92 | 0.94 | 0.93 | 2253 |
| 2 | 0.94 | 0.91 | 0.93 | 2140 |

Accuracy: 0.94

| VGG-16 | | | | |
|---|---|---|---|---|
| Class | Precision | Recall | f1-score | Support |
| 0 | 0.99 | 0.92 | 0.95 | 2395 |
| 1 | 0.91 | 0.96 | 0.93 | 2253 |
| 2 | 0.92 | 0.94 | 0.93 | 2140 |

Accuracy: 0.94

Table 6: Test-set performance metrics of pre-trained ImageNet models fine-tuned on the COVID-QU-Ex dataset.

## A.3 Ablation Study

| AlexNet | | | | |
|---|---|---|---|---|
| Class | Precision | Recall | f1-score | Support |
| 0 | 0.97 | 0.98 | 0.97 | 302 |
| 1 | 0.98 | 0.97 | 0.97 | 298 |

Accuracy: 0.97

| MobileNet | | | | |
|---|---|---|---|---|
| Class | Precision | Recall | f1-score | Support |
| 0 | 0.99 | 0.99 | 0.99 | 302 |
| 1 | 0.99 | 0.99 | 0.99 | 298 |

Accuracy: 0.99

| VGG-16 | | | | |
|---|---|---|---|---|
| Class | Precision | Recall | f1-score | Support |
| 0 | 1.00 | 0.97 | 0.99 | 302 |
| 1 | 0.97 | 1.00 | 0.99 | 298 |

Accuracy: 0.99

Table 7: Test-set performance metrics of pre-trained ImageNet models fine-tuned on the Br35h dataset.

## A.4 Adversarial Training

In this section, we report the final performance metrics of the adversarially trained DNN classifiers. We follow a similar training process as used for the original fine-tuned classifiers and train an ImageNet pre-trained model using a batch-size of 32, and a learning-rate of $1e-4$ using an ADAM optimiser Kingma & Ba (2015) and a Cross Entropy Loss. We augment each batch by adding random perturbations onto the images, resulting in a final batch-size of 64 consisting of both benign and its perturbed counterparts. We generate a random perturbation by first uniformly sampling the parameters $N$ and Maximum Diameter, then randomly sampling the characteristics of the perturbation as outlined in Section 3. We provide the pseudo-code of the procedure in Algorithm 1.

---

**Algorithm 1:** Adversarial Procedure employing EvoAttack perturbations

---

**1 Input:** Number of circles grid $N$, grid of Maximum diameters **MD**, number of epochs $T$, the batch-size $BS$, the learning rate $lr$, the loss function $L$, perturbation constraint $\epsilon$, classifier $f$

**2** Initialise $\theta$ // Initialise weights

**3 for** $t \leftarrow 1; t < T; t \leftarrow t+1$ **do**

**4**    **for** $j \leftarrow 1; j < BS; i \leftarrow BS+1$ **do**

**5**       $N \leftarrow \mathcal{U}(\boldsymbol{N})$ // sample number of circles

**6**       Max diameter $\leftarrow \mathcal{U}(\mathbf{MD})$ // sample maximum diameter

**7**       $\boldsymbol{\delta} \leftarrow \mathcal{U}(0,1)^{N \times 7}$ // random

**8**       $\mathbf{x}_{tj}^* \leftarrow CONSTRUCT(\mathbf{x}_j, \boldsymbol{\delta}, \epsilon)$ // Construct adversarial image (see Section 3.3)

**9**    $\theta \leftarrow \theta - \nabla_\theta L(f([\mathbf{x}_t, \mathbf{x}_t^*], [y, y])$ // Update mode weights with some optimizer, e.g. ADAM

---

### A.4.1 Adversarial Training Metrics

In this section, we report the final performance metrics of the trained DNN classifiers. As outlined in Section 4. The evaluation metrics of these models can be found in Tables 8,9 and 10.

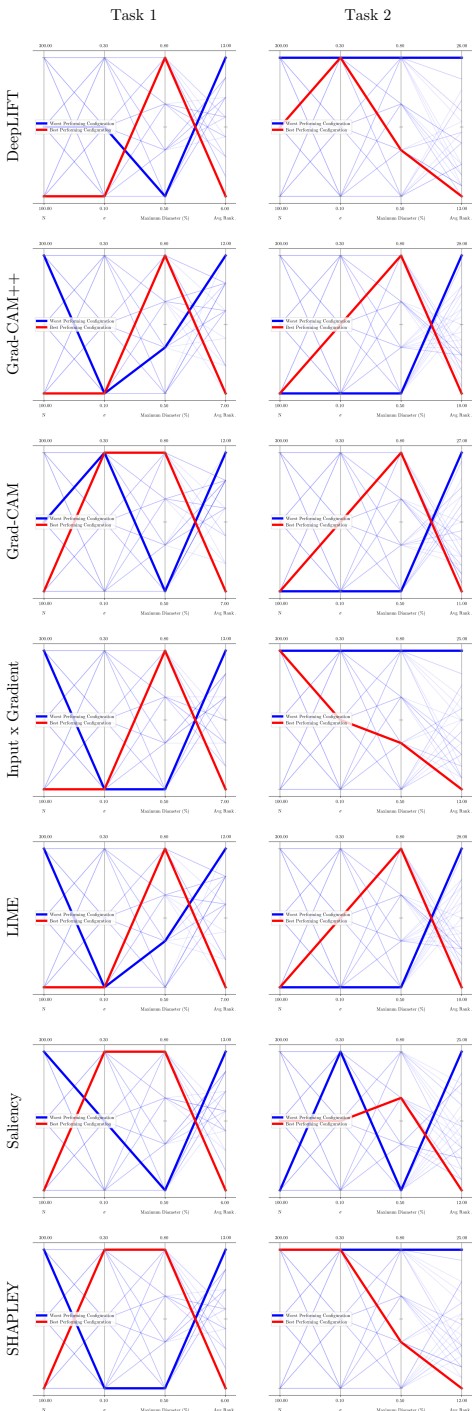

Figure 7: Parallel plots showing the results of configurations testing during the grid search. For each task we use its respective domination relation (described in Section 3.4) to rank its the performance of each configuration when attacking an image. Here we use the average rank (last column) to describe the performance of the configuration, where lower ranks describe better performing configurations. We highlight the best and worst performing configurations in blue and red, respectively.

| AlexNet | | | | |
|---|---|---|---|---|
| Class | Precision | Recall | f1-score | Support |
| 0 | 0.66 | 0.73 | 0.68 | 65 |
| 1 | 0.79 | 0.77 | 0.78 | 109 |
| 2 | 0.66 | 0.73 | 0.69 | 215 |
| 3 | 0.58 | 0.72 | 0.55 | 25 |
| 4 | 0.50 | 0.81 | 0.60 | 206 |
| 5 | 0.96 | 0.85 | 0.90 | 1354 |
| 6 | 0.87 | 0.90 | 0.88 | 29 |

Accuracy: 0.82

| MobileNet | | | | |
|---|---|---|---|---|
| Class | Precision | Recall | f1-score | Support |
| 0 | 0.72 | 0.69 | 0.66 | 65 |
| 1 | 0.78 | 0.85 | 0.81 | 109 |
| 2 | 0.77 | 0.71 | 0.74 | 215 |
| 3 | 0.85 | 0.68 | 0.76 | 25 |
| 4 | 0.77 | 0.73 | 0.67 | 206 |
| 5 | 0.94 | 0.93 | 0.93 | 1354 |
| 6 | 0.96 | 0.93 | 0.95 | 29 |

Accuracy: 0.87

| VGG-16 | | | | |
|---|---|---|---|---|
| Class | Precision | Recall | f1-score | Support |
| 0 | 0.74 | 0.75 | 0.75 | 65 |
| 1 | 0.87 | 0.83 | 0.85 | 109 |
| 2 | 0.76 | 0.78 | 0.77 | 215 |
| 3 | 0.76 | 0.52 | 0.62 | 25 |
| 4 | 0.69 | 0.69 | 0.69 | 206 |
| 5 | 0.94 | 0.94 | 0.94 | 1354 |
| 6 | 0.96 | 0.86 | 0.91 | 29 |

Accuracy: 0.88

Table 8: Test-set performance metrics of pre-trained ImageNet models adversarially trained on the HAM1000 dataset.

| AlexNet | | | | |
|---|---|---|---|---|
| Class | Precision | Recall | f1-score | Support |
| 0 | 0.96 | 0.91 | 0.94 | 2395 |
| 1 | 0.83 | 0.94 | 0.91 | 2253 |
| 2 | 0.90 | 0.84 | 0.89 | 2140 |

Accuracy: 0.90

| MobileNet | | | | |
|---|---|---|---|---|
| Class | Precision | Recall | f1-score | Support |
| 0 | 0.96 | 0.97 | 0.97 | 2395 |
| 1 | 0.87 | 0.94 | 0.93 | 2253 |
| 2 | 0.94 | 0.85 | 0.93 | 2140 |

Accuracy: 0.93

| VGG-16 | | | | |
|---|---|---|---|---|
| Class | Precision | Recall | f1-score | Support |
| 0 | 0.99 | 0.95 | 0.97 | 2395 |
| 1 | 0.89 | 0.97 | 0.93 | 2253 |
| 2 | 0.94 | 0.89 | 0.91 | 2140 |

Accuracy: 0.94

Table 9: Test-set performance metrics of pre-trained ImageNet models adversarially trained on the COVID-QU-Ex dataset.

## A.5 Proposed Method

| MobileNet | | | | |
|---|---|---|---|---|
| Class | Precision | Recall | f1-score | Support |
| 0 | 0.99 | 0.99 | 0.99 | 302 |
| 1 | 0.99 | 0.99 | 0.99 | 298 |

Accuracy: 0.99

| AlexNet | | | | |
|---|---|---|---|---|
| Class | Precision | Recall | f1-score | Support |
| 0 | 0.99 | 0.98 | 0.99 | 302 |
| 1 | 0.98 | 0.99 | 0.99 | 298 |

Accuracy: 0.99

| VGG-16 | | | | |
|---|---|---|---|---|
| Class | Precision | Recall | f1-score | Support |
| 0 | 0.98 | 0.97 | 0.98 | 302 |
| 1 | 0.97 | 0.98 | 0.98 | 298 |

Accuracy: 0.99

Table 10: Test-set performance metrics of pre-trained ImageNet models adversarially trained on the Br35h dataset.

---

**Algorithm 2:** Evolutionary Strategy for Generated Adversarial Images Against Explainable Methods

---

**1** **Input:** Explanation method $g$, trained DNN classifier $f$, distance measure $\mathcal{D}$, query budget $K$, number of circles $N$, evolutionary step-size $\sigma$, benign image $\mathbf{x}$

**2** $\boldsymbol{\delta} \leftarrow \mathcal{U}(0,1)^{N \times 7}$

**3** $\mathbf{x}^* \leftarrow CONSTRUCT(\mathbf{x}, \boldsymbol{\delta}^*, \epsilon)$ // see Section 3.3

**4** $D \leftarrow \mathcal{D}(g(\mathbf{x}^*), g(\mathbf{x}))$ // Measure explanation distance

**5** $y_{\mathbf{x}^*} \leftarrow \underset{p \in \{1, \cdots, P\}}{\operatorname{argmax}} f_p(\mathbf{x}^*)$

**6** **for** $k \leftarrow 1; k < K; k \leftarrow k + 1$ **do**

**7** $\quad$ $\boldsymbol{\delta}^{**} \leftarrow \boldsymbol{\delta} + \sigma \cdot \mathcal{N}(0, I)^{N \times 7}$ // Randomly adjust properties

**8** $\quad$ $\mathbf{x}^{**} \leftarrow CONSTRUCT(\mathbf{x}, \boldsymbol{\delta}^{**}, \epsilon)$

**9** $\quad$ $y_{\mathbf{x}^{**}} \leftarrow \underset{p \in \{1, \cdots, P\}}{\operatorname{argmax}} f_p(\mathbf{x}^{**})$

**10** $\quad$ $D^{**} \leftarrow \mathcal{D}(g(\mathbf{x}^{**}), g(\mathbf{x}))$

**11** $\quad$ $y_{\mathbf{x}^{**}} = y$ $D^{**} > D$ $\boldsymbol{\delta} \leftarrow \boldsymbol{\delta}^{**}$

**12** $\quad$ $\mathbf{x}^* \leftarrow \mathbf{x}^{**}$

**13** $\quad$ $y^* \leftarrow y^{**}$

**14** **return** $\boldsymbol{\delta}, \mathbf{x}^*, D$

---

### A.6 Qualitative Comparison

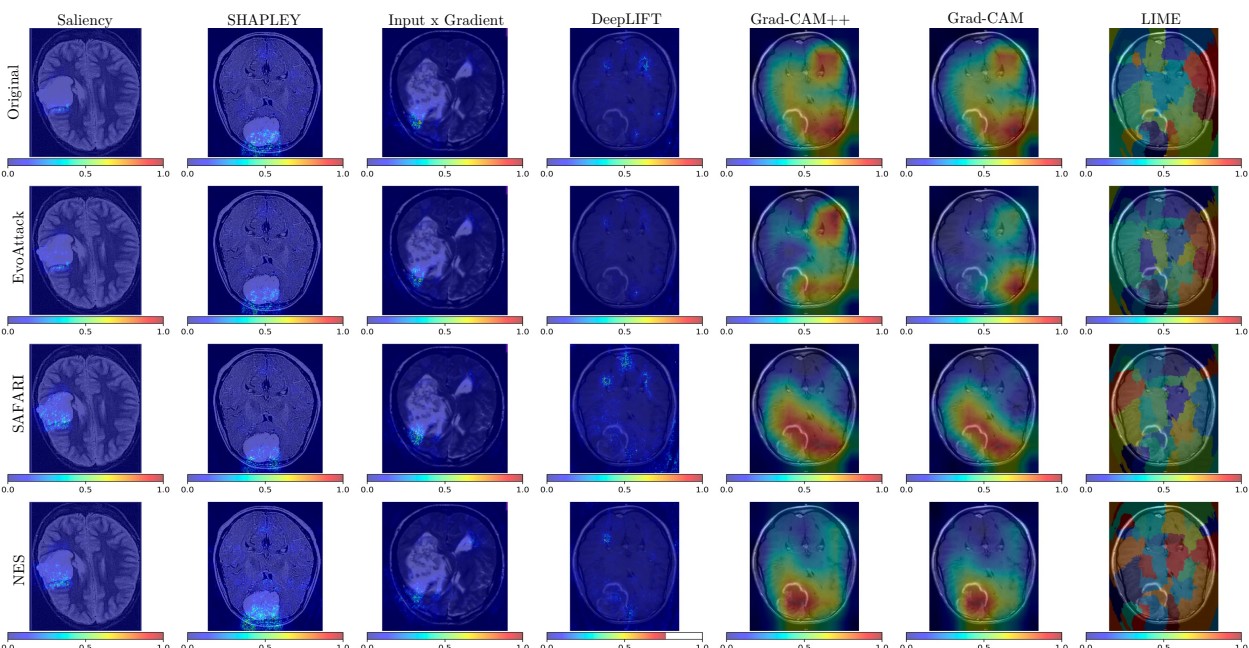

Figure 8: Original Br35h and adversarial images constructed by the proposed EvoAttack, SAFARI and NES attacks, along with attribution maps generated by the respective XAI method. Adversarial images are generated by attacks deployed within the Task 1 scenario. For the majority of the images, the proposed method is able to generate explanations similar to the original explanation, whereas explanations on SAFARI and NES generated adversarial images show larger distortions.

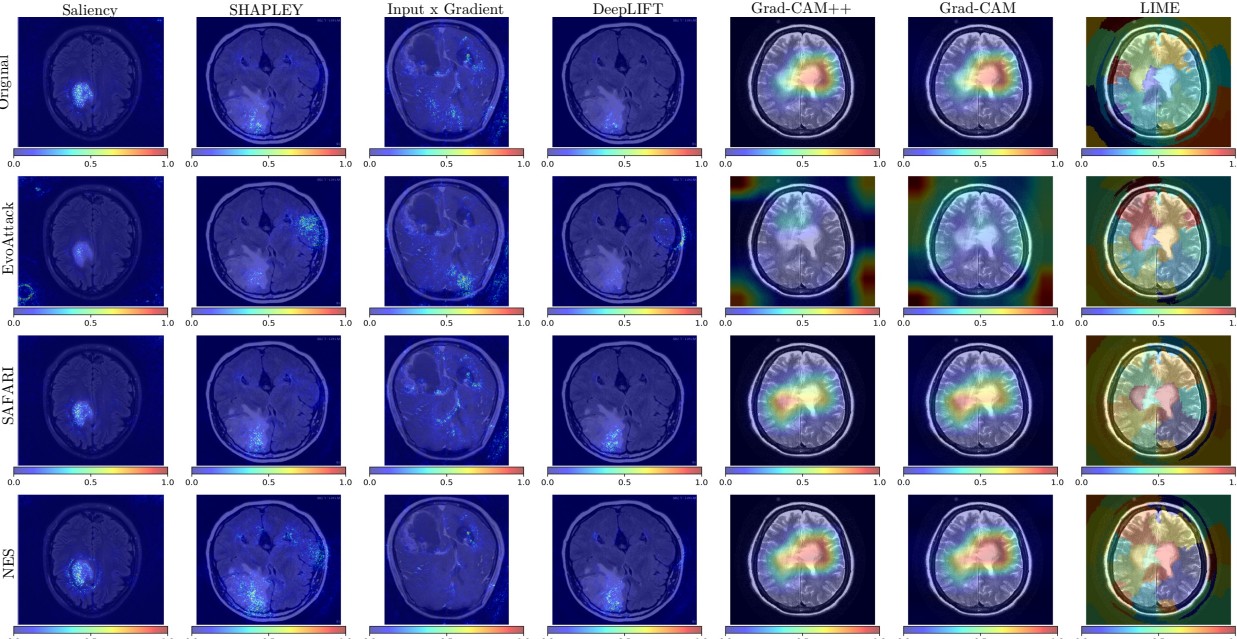

Figure 9: Original Br35h and adversarial images constructed by the proposed EvoAttack, SAFARI and NES attacks, along with attribution maps generated by the respective XAI method. Adversarial images are generated by attacks deployed within the Task 2 scenario. For the majority of the images, the proposed method is able cause larger distortions to the original explanation, compared to explanations on SAFARI and NES generated adversarial images.

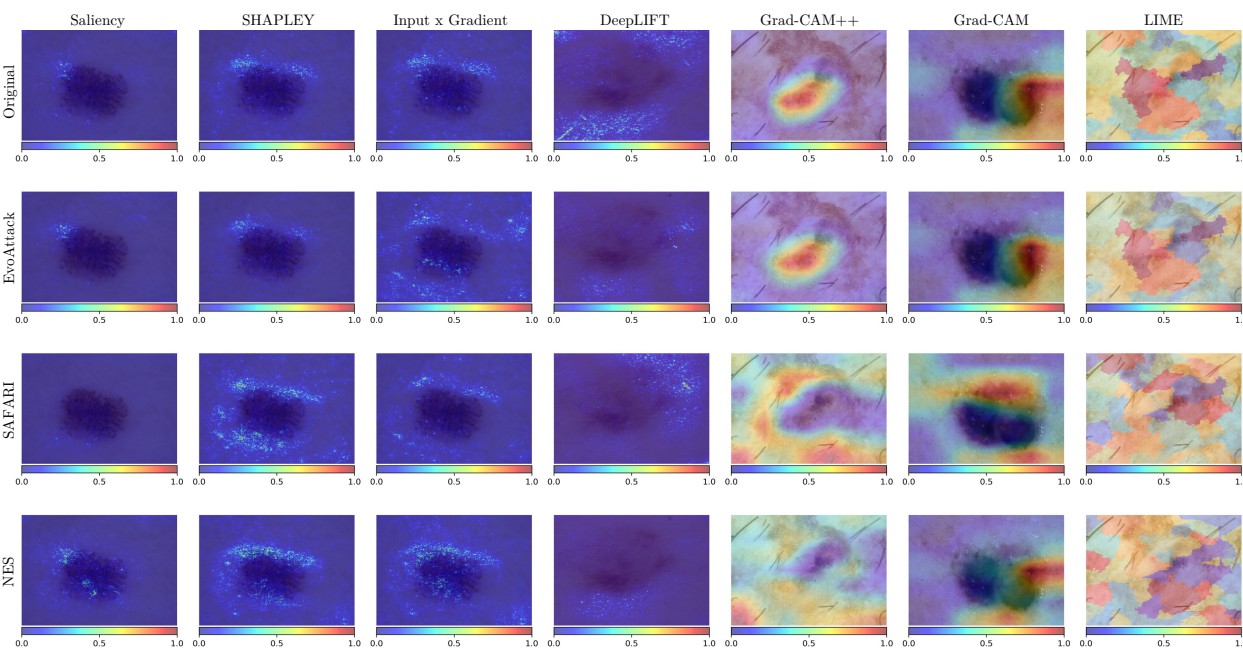

Figure 10: Original HAM10000 and adversarial images constructed by the proposed EvoAttack, SAFARI and NES attacks, along with attribution maps generated by the respective XAI method. Adversarial images are generated by attacks deployed within the Task 1 scenario. For the majority of the images, the proposed method is able to generate explanations similar to the original explanation, whereas explanations on SAFARI and NES generated adversarial images show larger distortions.

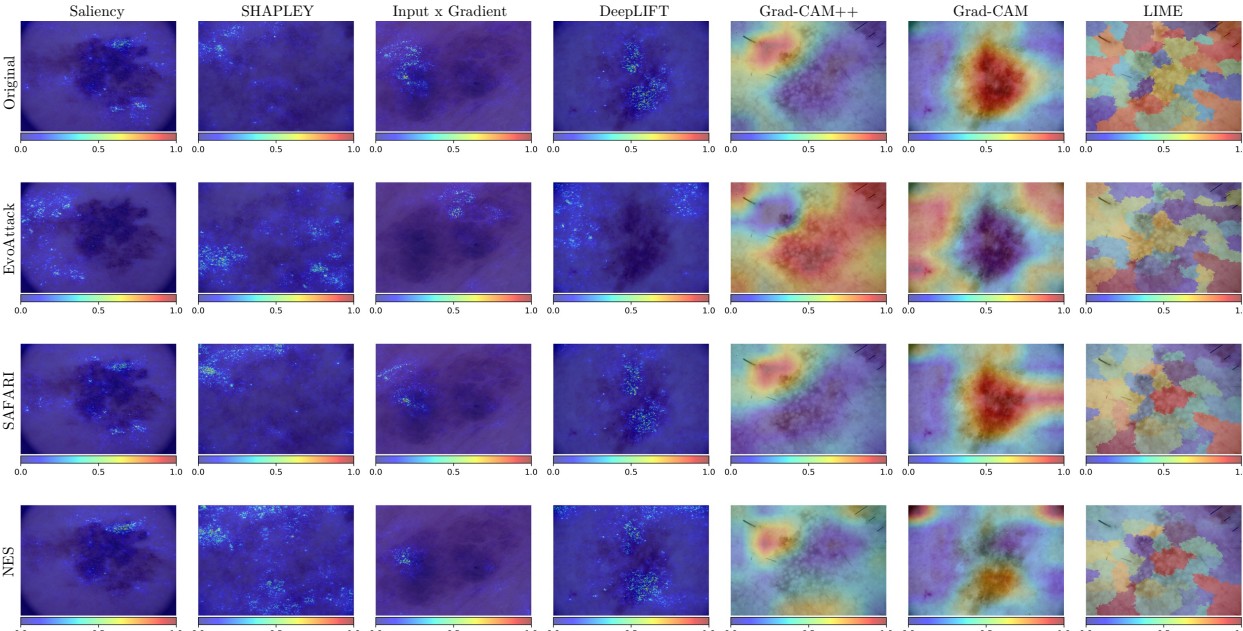

Figure 11: Original HAM10000 and adversarial images constructed by the proposed EvoAttack, SAFARI and NES attacks, along with attribution maps generated by the respective XAI method. Adversarial images are generated by attacks deployed within the Task 2 scenario. For the majority of the images, the proposed method is able cause larger distortions to the original explanation, compared to explanations on SAFARI and NES generated adversarial images.

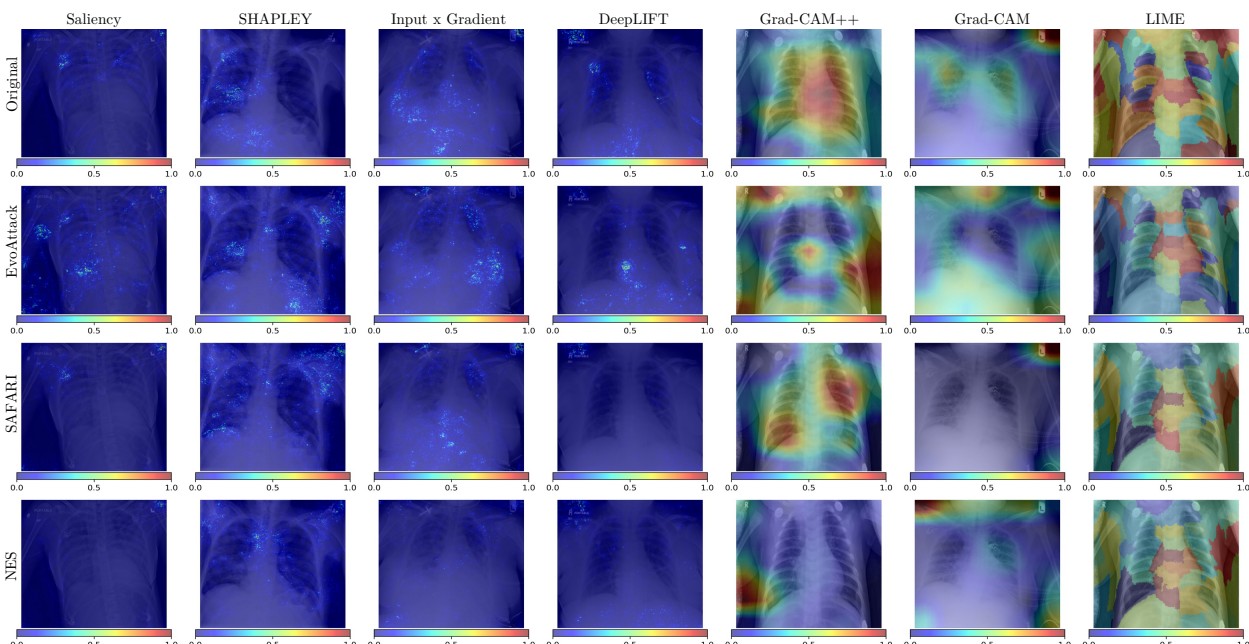

Figure 12: Original COVID-QU-Ex and adversarial images constructed by the proposed EvoAttack, SAFARI and NES attacks, along with attribution maps generated by the respective XAI method. Adversarial images are generated by attacks deployed within the Task 2 scenario. For the majority of the images, the proposed method is able to generate explanations similar to the original explanation, whereas explanations on SAFARI and NES generated adversarial images show larger distortions.

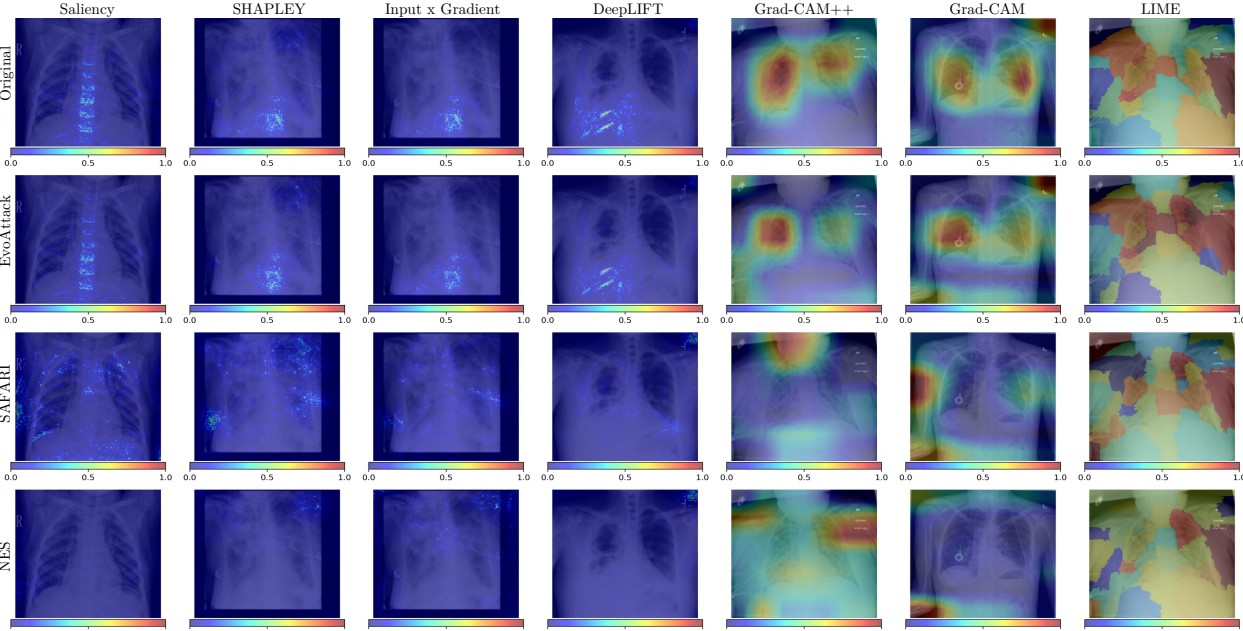

Figure 13: Original COVID-QU-Ex and adversarial images constructed by the proposed EvoAttack, SAFARI and NES attacks, along with attribution maps generated by the respective XAI method. Adversarial images are generated by attacks deployed within the Task 1 scenario. For the majority of the images, the proposed method is able cause larger distortions to the original explanation, compared to explanations on SAFARI and NES generated adversarial images.

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

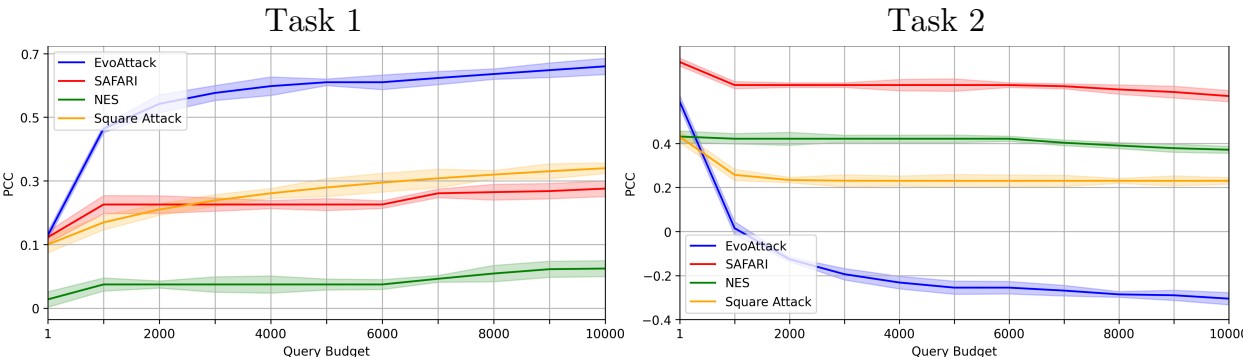

Figure 14: Mean PCC values with variance of attack methods when applied to task 1 (left) and task 2 (right) objectives for varying query budgets. The PCC metric is calculated from adversarial images that satisfy the respective task constraint and is computed by aggregating over all model architectures, explanation methods and datasets.

## A.8 Query Efficiency

To further demonstrate the effectiveness of the proposed EvoAttack, we conduct additional experiments for $K \in [1, 2000, 3000, 4000, 5000, 6000, 7000, 8000, 9000, 10000]$. The results in Figure 14 reveal that our method distorts attribution maps (as reflected in changing PCC values) much more rapidly than NES, SAFARI, or Square-Attack. This advantage stems from our method's ability to efficiently handle the high-dimensional perturbation search space as well as its adaptability to different granular XAI methods. EvoAttack reduces dimensionality by representing the perturbation as a set of overlapping circular shapes, which makes it invariant to image size. In contrast, NES and SAFARI directly optimize pixel-wise perturbations, causing them to become trapped in local optima—a common limitation of gradient- and heuristic-based strategies on high-dimension problems. Square-Attack shares some similarity with our approach in constructing shape-based perturbations, but its design limits its effectiveness for explanation distortion.

Table 15: Table presents the Pearson Correlation Coefficient (PCC) along with the percentage of images that satisfy the respective task constraint when attacking defended classifiers and XAI methods trained on the HAM10000, Br35h, COVID-QU-Ex and ImageNet datasets. We provide the mean and variance of each metric over 10 runs. Here adversarial training refers to the approach outlines in Section 4.4, Random Noise refers to the method of Qin et al. and Random Smoothing refers to the defence of Cohen et al..

| | Task 1 | | Task 2 | |
|---|---|---|---|---|
| Defence | Constraint Satisfied (↑) | PCC (↑) | Constraint Satisfied (↑) | PCC (↓) |
| Adversarial Training | $83.33\%(3.17)^{\ddagger}$ | $\mathbf{0.62(0.005)}^{\dagger}$ | $\mathbf{71.45\%(14.14)}^{\dagger}$ | $\mathbf{0.10(0.025)}^{\dagger}$ |
| Random Noise | $78.32\%(1.92)$ | $0.87(0.089)^{\ddagger}$ | $93.62\%(8.20)$ | $-0.03(0.301)^{\ddagger}$ |
| Random Smoothing | $\mathbf{77.93\%(1.27)}$ | $0.83(0.010)^{\ddagger}$ | $91.72(10.19)$ | $0.0(0.092)^{\ddagger}$ |

$^{\dagger}$ denotes the performance of the method significantly outperforms the compared methods according to the Wilcoxon signed-rank test Wilcoxon (1992) at the 5% significance level; $^{\ddagger}$ denotes the corresponding method is significantly outperformed by the best performing method (shaded).

## A.9   Comparison of Adversarial Defence Methods

The results in Table 14 compare the effectiveness of the proposed attack against three defense methods. We observe that the randomization-based approaches of Qin et al. and Cohen et al. provide greater resistance to adversarial classification in Task 1. This may be because both methods introduce uncertainty into the task constraint objective, thereby increasing the difficulty of inducing misclassification. However, once a successful adversarial perturbation is found, the adversarially trained model offers substantially stronger resistance. This outcome is expected, as randomization and smoothing defenses do not provide protection against explanation distortion once the classification constraint has been satisfied.

For Task 2, Table 14 shows that adversarial training offers more resistance to the proposed attack compared to the randomization-based defenses. This is likely due to the nature of the task: Task 2 requires that the image remain correctly classified. Since the primary goal of the randomization defenses is to prevent misclassification, ensuring correct classification is comparatively trivial.

## A.10 Comparison of Attack Speed

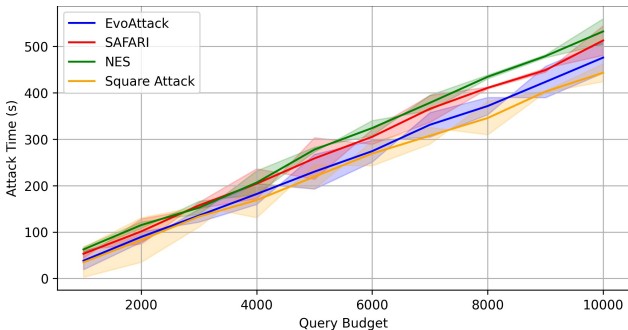

Figure 15: The figure shows the differing attack speeds (time to complete an attack) across different query budgets $K$. We see the speed of attack between methods remain similar. This can be expected as we assume the inference time of the attacked classifier being the largest factor of attack speed.

## A.11 Black-Box Adversarial Attacks

Despite the proposed method focusing on adversarial attacks against explainable methods, it can also be directly applied to black-box adversarial attacks but focusing solely on satisfying the constraint in (1). The black-box scenario in adversarial example generation can be categorized based on the assumed information obtained from the attacked DNN when queried with an input image. In this paper, we focus on the assumption that the attacker has access to the probabilities associated with each class, known as score-based attacks. However, other works address more restricted scenarios where only the top $\hat{C}$ class probabilities are accessible to the attacker Ilyas et al. (2018). Additionally, some work consider situations where only the predicted label of the attacked DNN is available to the attacker, referred to as decision-based attacks Brendel et al. (2018).

Most existing black-box score-based attack methods can be categorized into three main groups: transfer-, gradient-, and heuristic-based methods. The following subsections discuss notable methods from each of these categories.

Transfer-based methods conduct white-box attacks upon a surrogate model of the targeted DNN, to which the generated adversarial examples can be transferred Papernot et al. (2016); Guo et al. (2019); Cheng et al. (2019). Although these approaches claim to be query efficient by leveraging additional knowledge, they do not account for the computational cost incurred by training such substitute models. This can be particularly demanding when tackling large datasets like ImageNet. Furthermore many existing works assume the architecture of the surrogate model is highly similar to the target DNN, allowing a high percentage of adversarial examples to be successfully transferred.

Gradient-estimation methods works make use of white-box gradient-based attack methods, replacing the true gradient with an estimation. They have been widely studied for adversarial example generation by leveraging predicted class probabilities from DNNs to construct a loss function. Finite differences and some off-the-shelf gradient-based algorithms are then used as the optimizer, e.g., Chen et al. (2017); Tu et al. (2019); Ilyas et al. (2018); Bhagoji et al. (2018); Uesato et al. (2018); Yu et al. (2025). For example, Chen et al. proposed a zeroth-order optimization (ZOO) algorithm that applies Adam Kingma & Ba (2015) to generate adversarial examples. However, due to the high dimensionality of the image space, these techniques often require a large number of queries to estimate the perturbation's gradient at each step, which may not be practical in realistic black-box scenarios.

Heuristic methods circumvent the issues associated with gradient estimation, some research has focused on gradient-free heuristics to search for effective perturbations, and have achieved improved performance over

gradient estimation methods when the number of DNN queries is restricted Alzantot et al. (2018); Qiu et al. (2021); Andriushchenko et al. (2020).