# OpenReview forum: "An Evolutionary Algorithm for Black-Box Adversarial Attack Against Explainable Methods"
_TMLR — Accepted by TMLR_

### Review · Reviewer_3pRH · 2025-08-25

**Summary Of Contributions:**

This paper presents a black-box adversarial attack method validated on explainability methods for medical imaging applications. Unlike related approaches, the proposed method, EvoAttack, distorts images using semi-transparent circles with parameters determined by an evolutionary algorithm, requiring a much smaller computational budget. EvoAttack consistently outperformed two other black-box adversarial methods, NES and SAFARI, on two tasks across a variety of explainability methods and datasets.

**Additional Comments:**

**Strengths**:
- The paper is very well-written and clearly organized. Visualizations help ease the reading experience, particularly for those like myself who are not experts in this area
- Prior work is thoroughly explored and used to contextualize both the problem setting and the proposed method
- The proposed method appears to be novel, uniquely drawing upon the computational art literature to provide a more efficient approach
- Experiments are thorough and described in detail. Multiple datasets and explainability methods are used to validate EvoAttack
- Given the fixed computational budget, EvoAttack considerably outperforms existing black-box attack methods in virtually all settings

**Weaknesses**:
- While the quantitative results are compelling, I would like to see performance across a range of “query”/computational budgets. E.g., perhaps EvoAttack only demonstrates such a margin of superiority in the low compute budget regime chosen; does this advantage disappear with larger budgets? As someone who is not deeply familiar with this field, I would be interested to see some practical comparison of time taken per attack (if applicable) across various compute budgets as well.
- Experiments only compare to 2 baseline methods and one problem domain (medical imaging). What about other black-box attack methods such as those listed in Table 2 here [1]? I appreciate that experiments were comprehensive, but the generalizability of the method could be strengthened with validation in domains other than medical imaging as well.

**References**

[1] Baniecki, Hubert, and Przemyslaw Biecek. "Adversarial attacks and defenses in explainable artificial intelligence: A survey." Information Fusion 107 (2024): 102303.

**Audience:**

Yes

**Audience Explanation:**

This submission would appeal to researchers who work on explainability and robustness of deep image recognition models.

**Broader Impact Concerns:**

I feel that the Broader Impacts statement is adequate.

**Claims And Evidence:**

Yes

**Claims Explanation:**

Generally yes, but experiments are limited to only (a) one choice of compute budget, (b) two baseline black-box attack methods, and (c) one problem domain (medical imaging). Including additional evidence addressing some or all of these would strengthen the claims made in the paper.

**Requested Changes:**

**Main feedback**
- Include performance metrics and some measure of speed/throughput across various budgets. It remains unclear whether the fixed budget in these experiments favors the proposed method. I’m aware that this is an advantage to EvoAttack regardless, but it would be useful to know how these black-box methods perform across various budgets
- If possible, include other relevant black-box attack methods or explain why they cannot be included
- If possible, include experiments on a natural image dataset. This is not strictly necessary, but would certainly improve the paper

**Minor feedback**
- Incorrect in-text citations in several places. E.g., “Ghorbani et al.Ghorbani et al. (2019)”. I believe the authors need to use \pcite for a proper parenthetical citation (i.e., when referencing a work such that the reference is not part of the sentence).
- Figure 5: The reader does not yet know the term “(1+1)-ES structure” at this stage in the paper. I would either remove this from the caption or briefly define it.
- Remove extra comma after Eq 1
- Some discussion or quantification of the computational demands of your method would be appreciated.
- As a reader not familiar with this space, is attack optimization done *per image*? What is a typical compute budget? Some brief discussion would be helpful
- “Figure 8” should read “Table 8”
- Remove extra period at end of section 4
- At the end of the first limitations paragraph, there are dangling citations that are their own sentence. Unclear where these belong

---

> ### Author Response · Authors · 2025-09-06
> **Responses on Attack Speed, Query Budgets, Square-Attack Comparison, and Clarifications**
>
> We thank the reviewer for their detailed feedback and positive assessment of our work. Our responses are as follows:
>
> **\# Speed Evaluation**: As requested, we report attack speeds in Section A.10 (Comparison of Attack Speed) of the revised appendix. The figure compares time per attack across different query budgets $K$. We observe similar runtimes across methods, which is expected since classifier inference dominates the attack time.
>
> **\#Varying Budgets**: In Section A.8 (Query Efficiency) of the revised Appendix, we show that EvoAttack distorts attribution maps (via PCC changes) much more rapidly than NES, SAFARI, and Square-Attack. This stems from EvoAttack’s shape-based perturbation representation, which reduces dimensionality. NES and SAFARI optimize each pixel independently, making them prone to getting stuck as locally optimal regions. Square-Attack also uses shape-like perturbations but is less effective, as shown in Tables 11–14 of the appendix. Due to text limitation, for further details, we refer the reviewer to our **\#Varying Budgets** discussion in the response to Reviewer **jqnL**.
>
> **\#Additional Black-box attack**: We also compare against Square-Attack (ECCV 2020). Results are shown in **\#Additional Black-box Attack** of the global response, and dataset-specific results are in Tables 11–14 of the appendix. While Square-Attack satisfies constraints effectively, it distorts explanations less than EvoAttack. Likely reasons include: (i) its restricted perturbation space ([$-\epsilon$, $+\epsilon$]); (ii) sequential square placement, which can lead to sub-optimal convergence; and (iii) iterative square shrinking, which limits its impact on both coarse- and fine-grained explanation methods. Due to text limitations, we refer the reviewer to **\#Varying Budgets** in our response to Reviewer **jqnL** for further discussion.
>
> **Table [1] Attacks**: The table refered to by the reviewer refers summarizes attacks across various modalities and settings. Our work focuses on image (modality I) and black-box (model B) attacks, where the adversary has access only to classifier output probabilities and XAI attribution maps. We compare against all methods within this setting shown in the table.
>
> **\#Natural Images**: As requested, we extend experiments to directly attack ImageNet images. Results across models and explanation methods are provided in **\#ImageNet** of our global response, where EvoAttack consistently outperforms comparative attacks.
>
> **\# Per image definition**: To improve clarity, we revised Section 3.1 (Problem Formulation) to state explicitly that EvoAttack operates per image:
> *“Consider a trained DNN classifier $f:\mathcal{X} \subseteq [0,1]^{h \times w \times 3} \rightarrow \mathbb{R}^{P}$ which takes a single benign RGB image we wish to attack $\mathbf{x}\in\mathcal{X}$ of height $h$ and width $w$, and outputs a label $y = \underset{p\in\{1,\cdots,P\}}{argmax}\ f_p(\mathbf{x})$, where $P$ is the total number of class labels.”*
>
> **\#Typos, grammer and captions**: We thank the reviewer for noting these issues. We carefully proofread the manuscript and have corrected the highlighted grammatical errors, and applied the reviewer’s suggestions. For example, we removed “(1+1)-ES” from the specified figure caption.

---

### Review · Reviewer_62Tr · 2025-08-27

**Summary Of Contributions:**

The manuscript examines the vulnerability of XAI methods in computer vision. Recent studies reveal these explanations can be manipulated by adversarial perturbations. Existing attacks, particularly white-box methods, lack generalizability, while black-box approaches often rely on costly meta-heuristics with high query demands. Moreover, current techniques overlook the varying granularity of XAI methods, performing poorly against region-based explanations such as Grad-CAM. To address these gaps, the authors propose a shape-based black-box attack inspired by computational art. By optimizing semi-transparent RGB shapes with evolutionary strategies, the method reduces the search space, operates efficiently under limited queries, and adapts to different explanation granularities. Evaluations demonstrate its effectiveness across diverse XAI methods.

**Audience:**

Yes

**Audience Explanation:**

The work directly addresses the intersection of XAI, adversarial robustness, and evolutionary optimization, all of which are active research areas in the machine learning community. By proposing a novel and query-efficient black-box attack EvoAttack that effectively compromises multiple explanation methods, the paper contributes to both the theory and practice of XAI robustness evaluation.

**Broader Impact Concerns:**

Despite its close connection to the black-box optimization setting, the paper provides very little discussion of zeroth-order optimization techniques. These methods are widely studied for query-limited adversarial attacks, and it would be valuable to clarify how the proposed shape-based evolutionary strategy differs from, or improves upon, established zeroth-order approaches. Without such discussion, readers may find it difficult to fully situate this work in the broader landscape of black-box adversarial optimization. For example, prior works such as Yu et al. (2025, IEEE TKDE) on GZOO and Chen et al. (2017, AISec) on ZOO provide relevant baselines and conceptual context.

1. Yu et al. GZOO: Black-Box Node Injection Attack on Graph Neural Networks via Zeroth-Order Optimization. IEEE TKDE 2025.
2. Chen et al. ZOO: Zeroth Order Optimization based Black-box Attacks to Deep Neural Networks without Training Substitute Models. AISec 2017.

**Claims And Evidence:**

Yes

**Claims Explanation:**

- The method is evaluated across three diverse medical imaging datasets (HAM10000, Br35h, COVID-QU-Ex), using multiple widely adopted classifiers (MobileNet, AlexNet, VGG-16) and seven explanation methods (e.g., Grad-CAM, SHAPLEY, Saliency).
- Extensive results are presented, including constraint satisfaction rates and PCC scores, which clearly demonstrate EvoAttack’s superior performance compared to state-of-the-art methods such as NES and SAFARI. Statistical validation with the Wilcoxon signed-rank test further confirms the significance of improvements.
- Visual comparisons of adversarial images and attribution maps (Figures 3, 4, 7) convincingly illustrate how EvoAttack achieves either subtle misclassification with preserved explanations or distorted explanations with preserved classifications.

**Requested Changes:**

- The experiments are conducted solely on medical imaging datasets (HAM10000, Br35h, COVID-QU-Ex). Although this domain is highly relevant and safety-critical, it is not clear whether the proposed attack generalizes to other computer vision tasks, such as natural images or large-scale benchmarks (e.g., ImageNet). Broader evaluation would strengthen the claim of general applicability.
- The comparisons focus primarily on NES and SAFARI. While these are appropriate black-box adversarial baselines, the exclusion of other query-efficient or evolutionary optimization attacks (e.g., Square Attack, genetic search methods) may leave some doubt about whether the performance gains are universally superior.
- The attack assumes access to both the DNN predictions and the outputs of the XAI method. In practice, this dual access may not always be available in real-world settings, particularly in sensitive domains like healthcare. Clarifying the threat model and its practical feasibility would improve the paper’s impact.
- The exploration of defenses is somewhat limited. The study considers adversarial training, which is a natural baseline, but does not evaluate other plausible defense strategies such as noise injection, explanation regularization, or randomized smoothing. This limits the depth of the defense discussion.

---

> ### Author Response · Authors · 2025-09-06
> **Responses on Natural Images, Additional Attacks, Threat Model, Defenses, and Zeroth-Order Attacks**
>
> We thank the reviewer for their detailed feedback and their positive comments on our experiments and visualizations. We address the concerns as follows:
>
> **\#Natural Images**:
> In the original work, we used ImageNet images to optimize the parameters of the proposed attack (Section 4.5, ablation study). As requested, we extend our experiments by directly attacking ImageNet images. Results across all models and explanation methods are presented in **\#ImageNet** of our global response, where EvoAttack consistently outperforms competing methods.
>
> **\#Additional Black-box attack**: We additionally compare against Square-Attack, a state-of-the-art method for black-box attacks on classifiers. Results are shown in **\#Additional Black-box Attack** of our global response, and dataset-specific results are provided in Tables 11–14 of the revised appendix. While Square-Attack satisfies task constraints effectively, it distorts explanation methods far less than EvoAttack. Likely reasons include: (i) its restricted perturbation search space ([$-\epsilon$, $+\epsilon$]); (ii) its sequential square placement, which makes it prone to sub-optimal convergence; and (iii) its iterative shrinking strategy, which reduces effectiveness against both coarse- and granular explanation methods. Due to text limitations, we refer the reviewer to **\#Varying Budgets** in our response to Reviewer **jqnL** for further discussion.
>
> **\#Threat clarification**: We revised Section 3.1 (Problem Formulation) to clarify assumptions. Specifically, we state:
> “*In this work, we assume access to the output probabilities of the classifier $f$, and attack an XAI method $g$ that outputs an attribution map $g(\cdot, \cdot) \rightarrow \mathbb{R}^{h \times w}$, where its height and width match the input image $\mathbf{x}$.*”
> We also amended the conclusion to highlight this assumption and suggest direction of future research that extends decision-only attacks (where the attack only has access to the final classification label of an input) for robustness evaluation of explanation methods.
>
> **\#Defence comparison**: As shown in **\#Defence Comparisons** of our global response, we evaluated EvoAttack against adversarial training, Random Noise Defense, and Randomized Smoothing. We observe that random defense methods pose greater resistance in satisfying the Task 1 constraint by introducing uncertainty into the optimization objective, but adversarial training ultimately provides stronger resistance once valid perturbations are found. This is expected, since randomization does not protect against explanation distortion. In Task 2, adversarial training again provides greater robustness, while random defenses are less effective because the task constraint requires maintaining correct classification, which is outside their intended scope. Results and further discussion are provided in Section A.9 of the revised appendix (highlighted in blue).
>
>
> **\#Zeroth-order attacks**: To strengthen the discussion of black-box adversarial attacks, we add a new subsection (A.11 Black-Box Adversarial Attacks) to the appendix. For zeroth-order optimization, we include the following text:
> *“Gradient-estimation methods adapt white-box gradient-based attacks by replacing the true gradient with an estimate. These approaches construct a loss function using predicted class probabilities and estimate its gradient using finite differences, applying algorithms such as ADAM for its optimization [1, 2, 3]. Despite their success in generating adversarial examples against deep neural networks, the large computational budget for computing gradients across all pixels makes them limited within query-restricted scenarios.”*
>
> References:
> [1] Chen et al. ZOO: Zeroth Order Optimization based Black-box Attacks to Deep Neural Networks without Training Substitute Models. AISec 2017.
> [2] Tu et al. Autozoom: Autoencoder-based zeroth order optimization method for attacking black-box neural networks. AAAI 2019.
> [3] Yu et al. GZOO: Black-Box Node Injection Attack on Graph Neural Networks via Zeroth-Order Optimization. IEEE TKDE 2025.

---

### Review · Reviewer_jqnL · 2025-08-27

**Summary Of Contributions:**

The paper proposes EvoAttack, a black-box adversarial attack towards Explainable AI (XAI) in the context of medical imaging. The authors use the semi-transparent, RGB-valued circles to create perturbations, which largely reduce the search space of attacks, thus leading to higher attack effects with fewer query numbers. Finally, the paper discusses the vulnerabilities of different XAI methods based on the attack results.

**Additional Comments:**

Please see the requested changes.

**Audience:**

Yes

**Audience Explanation:**

The findings of the paper can be valuable to the community for a better understanding of the robustness of XAI methods in medical imaging. The discussion in the adversarial training can benefit future research, which tends to improve the robustness and reliability of the XAI methods.

**Claims And Evidence:**

Yes

**Claims Explanation:**

The paper provides comprehensive and extensive experiments to support each of the claims. The experiments are conducted across three different medical imaging datasets, including HAM10000, Br35h, and the COVID-QU-Ex dataset; seven XAI methods; and three different classifier networks, including MobileNet, AlexNet, and VGG. The experiments are repeated over 10 different random seeds and are reported with the final mean and variance.

**Requested Changes:**

Despite the strength mentioned above, I recommend addressing these issues before I make my recommendation for acceptance.

- Lots of typos. I find the paper has a lot of typos, repetition, and even incompleteness. For example, in the last second paragraph on page 4, the sentence is not complete. In Equation 1, the $\delta$ is not consistently bolded. There is an extra comma at the end of page 5. There is repetition in the paragraph above Section 4.2. Please carefully review the whole paper again and fix all the typos.
- Attack effectiveness versus query number. It's interesting to see the comparison between EvoAttack and other black-box attack approaches under different query numbers. For example, the x-axis represents the query number, and the y-axis represents the attack effectiveness.

---

> ### Author Response · Authors · 2025-09-06
> **Responses on Typos and Query Budget Analysis**
>
> We thank the reviewer for the positive feedback and constructive suggestions. Our responses are as follows:
>
> **\#Typos**: We appreciate the reviewer’s careful reading. All highlighted typos have been corrected, and we have conducted a thorough proofread of the revised manuscript.
>
> **\#Varying Budgets**: In the revised appendix (Section A.8, Query Efficiency), we include plots showing performance under varying query budgets. The results reveal that our method distorts attribution maps (as reflected in changing PCC values) much more rapidly than NES, SAFARI, or Square-Attack. This advantage stems from our method’s ability to efficiently handle the high-dimensional perturbation search space as well as its adaptability to different granular XAI methods. EvoAttack reduces dimensionality by representing the perturbation as a set of overlapping circular shapes, which makes it invariant to image size. In contrast, NES and SAFARI directly optimize pixel-wise perturbations, causing them to become trapped in local optima—a common limitation of gradient- and heuristic-based strategies on high-dimension problems.
>
> Square-Attack shares some similarity with our approach in constructing shape-based perturbations, but its design limits its effectiveness for explanation distortion, as demonstrated in Tables 11–14 of the appendix and **\# Additional Black-box Attack** of our global response. Specifically:
>
> Restricted search space: Square-Attack samples perturbations only on the constraint boundary ([$-\epsilon$, $+\epsilon$]), which is effective for classification attacks but overly restrictive for more complex tasks such as explanation distortion.
>
> Sequential square placement: After initially perturbing the full image by constructing the perturbation from vertical strips, it randomly adds one square at a time. Once accepted, a square cannot be removed, making the method prone to local optima. EvoAttack, by contrast, can adjust its perturbation to cover either a broad region or localized areas of the image.
>
> Iterative square shrinking: As the attack progresses, the square size decreases. For explanation methods focusing on less granular features (e.g. Grad-Cam), fine-grained perturbations have lesser impact. For more granular XAI methods, the sparse structure of Squre-Attack perturbations has less effectiveness. EvoAttack, however, can adapt to varying levels of granularity through its attack parameters.
>
> We added this discussion to the appendix of the revised manuscript (highlighted in blue for clarity).

---

### Author Response · Authors · 2025-09-06
**Summary of Additional Experiments, Clarifications, and Revisions**

We thank all reviewers for their constructive feedback, which has helped us improve the quality and clarity of our work. Several of the suggested experiments overlapped across reviews; to avoid redundancy and due to space limitations, we summarise them here and add any necessary results.

**\#ImageNet**: As requested, we evaluated our approach on ImageNet to further demonstrate its generalizability. Results are shown below.

**Table:** Pearson Correlation Coefficient (PCC) along with the percentage of images that satisfy the respective task constraint when attacking images from the ImageNet dataset. We provide the mean and variance of each metric over 10 runs.

| Attack        | Task 1: Constraint Satisfied (↑) | Task 1: PCC (↑) | Task 2: Constraint Satisfied (↑) | Task 2: PCC (↓) |
|---------------|---------------------------------|----------------|---------------------------------|----------------|
| Square-Attack | **81.84%(14.25)**               | 0.50(0.052)‡ | 84.85%(49.49)                   | 0.20(0.001)‡ |
| EvoAttack     | 81.10%(13.74)                    | **0.75(0.030)†** | **85.92%(51.41)**             | **-0.33(0.132)†** |
| NES           | 24.23%(0.274)‡               | 0.27(0.009)‡ | 78.16%(52.89)‡               | 0.44(0.013)‡ |
| SAFARI        | 19.05%(22.85)‡               | 0.48(0.052)‡ | 62.80%(74.85)‡               | 0.67(0.017)‡ |

**\# Additional Black-box Attack**:
We also conducted further comparisons against the state-of-the-art Square Attack (Andriushchenko et al., ECCV 2020), confirming the superior performance of our method (table below).

**Table:** Pearson Correlation Coefficient (PCC) and percentage of images satisfying constraints for ImageNet attacks.

**Table:** Pearson Correlation Coefficient (PCC) along with the percentage of images that satisfy the respective task constraint when attacking images from the HAM10000, Br35h, COVID-QU-Ex and ImageNet datasets. We provide the mean and variance of each metric over 10 runs.

| Method        | Constraint Satisfied (↑)   | PCC (↑)                    | Constraint Satisfied (↑)   | PCC (↓)                    |
|---------------|-----------------------------|-----------------------------|-----------------------------|-----------------------------|
| **Task 1**    |                             |                             | **Task 2**                  |                             |
| Square-Attack | **83.43% (2.379)**          | 0.454 (0.141)‡              | 81.65% (2.545)              | 0.283 (0.243)‡              |
| EvoAttack     | 82.23% (1.785)              | **0.770 (0.077)†**          | **82.80% (1.703)**          | **-0.224 (0.261)†**         |

**\#Typos and Additional discussion**:
We appreciate reviewers pointing the writing errors and missing clarifications. These have been corrected in the revised manuscript, with amended and additional text highlighted in blue.

**\#Varying Budgets**:
In response to reviewer suggestions, we ran additional experiments across all four datasets under varying query budgets. The results are reported in Section A.8 (Query Efficiency) of the Appendix.

**\# Defence Comparisons**:
We compared our method against two defense strategies—Random Noise Defense (Qin et al., NeurIPS 2021) and Randomized Smoothing (Cohen et al., ICML 2019)—in addition to adversarial training. Results are shown below.
**Table**:
| Defence              | Task 1: Constraint Satisfied (↑) | Task 1: PCC (↑)     | Task 2: Constraint Satisfied (↑) | Task 2: PCC (↓)      |
| -------------------- | -------------------------------- | ------------------- | -------------------------------- | -------------------- |
| Adversarial Training | 83.33% (3.17)‡                | **0.62(0.005)†** | **71.45% (14.14)†**           | **0.10 (0.025)†** |
| Random Noise         | 78.32% (1.92)                    | 0.87 (0.089)‡    | 93.62% (8.20)                    | -0.03 (0.301)‡    |
| Random Smoothing     | **77.93% (1.27)**                | 0.83(0.010)‡     | 91.72 (10.19)                    | 0.0 (0.092)‡      |

---

### Decision · Action_Editor_ghfi · 2025-10-02

**Recommendation:** Accept with minor revision

**Additional Comments:**

Please include the further experimental results in the author feedback stage to the final manuscript.

Also I recommend polishing the writing of the paper further including fixing typos and clarify explanations of the designed attack & experimental settings.

**Audience:**

Yes

**Audience Explanation:**

Machine learning researchers studying the (adversarial) robustness of neural networks to attacks. Also machine learning researchers interested in explainable AI (XAI).

**Claims And Evidence:**

Yes

**Claims Explanation:**

The paper proposes EvoAttack on Explainable AI (XAI) methods applied to medical imaging. The main contribution seems to be on the design space of the injected perturbation (semi-transparent, RGB-valued circles), and the claim is that this design reduces search space so that the number of optimisation steps for finding an adversarial example is decreased.

The paper provides a comprehensive set of experiments to support the claim.

During reviewing, reviewers mainly raised concerns regarding the technical details of the attack (e.g., the budget of attack) as well as stronger baselines (e.g., stronger defence methods and other attack baselines). Also they wonder whether the proposed method also works for image datasets beyond medical image scans.

In response, the authors provided further experimental results, which largely cleared reviewers' concerns.